



# Improved tropospheric and stratospheric sulfur cycle in the aerosol-chemistry-climate model SOCOL-AERv2

Aryeh Feinberg[1,2,3], Timofei Sukhodolov[1,4], Bei-Ping Luo[1], Eugene Rozanov[1,4], Lenny H. E. Winkel[2,3], Thomas Peter[1], and Andrea Stenke[1]

[1]Institute for Atmospheric and Climate Science, ETH Zurich, Zurich, Switzerland
[2]Institute of Biogeochemistry and Pollutant Dynamics, ETH Zurich, Zurich, Switzerland
[3]Eawag, Swiss Federal Institute of Aquatic Science and Technology, Dübendorf, Switzerland
[4]Physikalisch-Meteorologisches Observatorium Davos and World Radiation Center, Davos, Switzerland

**Correspondence:** Aryeh Feinberg (aryeh.feinberg@env.ethz.ch)

**Abstract.** SOCOL-AERv1 was developed as an aerosol-chemistry-climate model to study the stratospheric sulfur cycle and its influence on climate and the ozone layer. It includes a sectional aerosol model that tracks the sulfate particle size distribution in 40 size bins, between 0.39 nm to 3.2 μm. Sheng et al. (2015) showed that SOCOL-AERv1 successfully matched observable quantities related to stratospheric aerosol, including a simulated stratospheric aerosol burden of 109 Gg of sulfur (S), very close

to the satellite-derived estimate available in 2015, 112 Gg S. In the meantime, both the satellite retrieval and SOCOL-AER have undergone significant improvements. In producing SOCOL-AERv2 we have implemented several updates to the model: adding interactive deposition schemes, improving the sulfate mass and particle number conservation, and expanding the tropospheric chemistry scheme. We compare the two versions of the model with background stratospheric sulfate aerosol observations, stratospheric aerosol evolution after Pinatubo, and ground-based sulfur deposition networks. SOCOL-AERv2 shows similar

levels of agreement as SOCOL-AERv1 with satellite-measured extinctions and in situ optical particle counter (OPC) balloon flights. Also, the volcanically quiescent total stratospheric aerosol burden simulated in SOCOL-AERv2, 160 Gg S, agrees very well with the new satellite estimate of 165 Gg S. However, SOCOL-AERv2 simulates too high cross-tropopause transport of tropospheric $SO_2$ and/or sulfate aerosol, leading to an overestimation of lower stratospheric aerosol. Due to the current lack of upper tropospheric $SO_2$ measurements and the neglect of organic aerosol in the model, the lower stratospheric bias of

SOCOL-AERv2 was not further improved. Model performance under volcanically perturbed conditions has also undergone some changes, resulting in a slightly lower shorter volcanic aerosol lifetime after the Pinatubo eruption. With the improved deposition schemes of SOCOL-AERv2, simulated sulfur wet deposition fluxes are within a factor of 2 of measured deposition fluxes at 78% of the measurement stations globally, an agreement which is on par with previous model intercomparison studies. Because of these improvements, SOCOL-AERv2 will be better suited to studying changes to atmospheric sulfur deposition

due to variations in climate and emissions.



# 1 Introduction

The atmospheric sulfur cycle is of significance for climate, atmospheric chemistry, ecosystems, agriculture, and human health. Sulfate aerosol reflects incoming shortwave solar radiation, leading to a cooling effect at the Earth's surface. Sulfate aerosol also absorbs outgoing longwave radiation, leading to a warming of the lower stratosphere. In addition to these direct radiative effects,

sulfate particles act as cloud condensation nuclei, leading to cloud formation and an indirect radiative effect (Myhre et al., 2013). Sulfate particles affect air chemistry in the troposphere and stratosphere by catalyzing chemical reactions that deactivate nitrogen (Dentener and Crutzen, 1993). In the cold polar winter stratosphere, they affect ozone depletion by activating chlorine species and serving as condensation nuclei for polar stratospheric clouds (Solomon, 1999). For decades, atmospheric sulfate deposition has been a concern due to its role in acidification of soils and surface waters (Vet et al., 2014). On the other hand,

sulfur is a macronutrient for plants, and decreasing sulfur deposition has led to increased demand for sulfur fertilizers in certain regions (Hinckley and Matson, 2011; Kovar and Grant, 2011). These wide-ranging impacts have motivated the development of atmospheric sulfur models.

Sulfur is emitted to the atmosphere in various compounds through both natural and anthropogenic sources. Natural sources of sulfur include $SO_2$ from eruptive and degassing volcanoes, and dimethylsulfide (DMS) from marine phytoplankton and in small

amounts from the terrestrial biosphere. Anthropogenic activities such as fossil fuel combustion, metal smelting, and biomass burning release sulfur mainly as $SO_2$ to the atmosphere (Smith et al., 2011). Short-lived sulfur compounds such as hydrogen sulfide ($H_2S$), DMS, and carbon disulfide ($CS_2$) are almost completely oxidized in the troposphere and thus do not enter the stratosphere in large amounts. Longer-lived sulfur compounds, such as carbonyl sulfide (OCS) and to some extent sulfur dioxide ($SO_2$), are transported to the stratosphere where they ultimately oxidize to gaseous sulfuric acid ($H_2SO_4$) (Thomason and Peter,

2006; Kremser et al., 2016). $H_2SO_4$ molecules nucleate to form new sulfate aerosol particles or condense on existing particles. In the stratosphere, this sustains a layer of binary $H_2SO_4$–$H_2O$ particles between 15 and 30 km, often called the "Junge layer" (Junge et al., 1961). The Junge layer is intensified compared to background conditions by sporadically occurring volcanic eruptions whose emissions reach the stratosphere. In the troposphere, $H_2SO_4$-containing particles are removed by wet and dry deposition, closing the atmospheric sulfur cycle (Kremser et al., 2016).

OCS was first suggested to be an important contributor to the stratospheric aerosol layer by Crutzen (1976). Recent modelling studies have quantified the contribution of different sulfur compounds to the stratospheric aerosol burden. Brühl et al. (2012) attributed 70% of the background, non-volcanic stratospheric aerosol burden above 20 km to OCS oxidation. Sheng et al. (2015) suggested that 56% of the stratospheric burden arises due to OCS and 32% due to $SO_2$ emissions. It must be noted that these studies calculated these contributions by turning off emissions from all other sulfur species, and lower sulfur emissions

can lead to smaller aerosol particles, slower sedimentation, and longer aerosol lifetimes.

As reviewed by Kremser et al. (2016), other studies have emphasized the minor, yet non-negligible, contribution of non-sulfate species to the stratospheric aerosol burden. Meteoritic dust particles enter the atmosphere at a rate of 3–300 t d$^{-1}$ (Plane, 2012), which would correspond to 0.15–15% of the stratospheric aerosol mass flux estimated by Sheng et al. (2015). Modelling (Yu et al., 2015) and measurements (Froyd et al., 2009; Friberg et al., 2014; Murphy et al., 2014) suggest that organic





carbon is a significant fraction of the aerosol mass in the lowest part of the stratosphere. The non-sulfate aerosol species are often not considered in stratospheric modelling studies, despite their possible contribution to observed aerosol quantities.

There are around fifteen active models that include stratospheric aerosol microphysics, which can be separated into models with sectional (∼1/3) or modal (∼2/3) size distributions (Kremser et al., 2016). SOCOL-AERv1 is a model with a sectional
scheme that divides the sulfate aerosol size distribution into 40 bins (Sheng et al., 2015). The model succeeded in reproducing the observed background stratospheric aerosol extinctions compared to the Stratospheric Aerosol and Gas Experiment II (SAGE II) and Halogen Occultation Experiment (HALOE) measurements (Thomason, 2012), as well as the particle size distributions measured by Optical Particle Counters (OPC) in the midlatitudes (Deshler et al., 2003; Deshler, 2008). The SOCOL-AERv1 simulated aerosol burden of 109 Gg sulfur (S) also matched the stratospheric burden calculated from SAGE-4$\lambda$ data
(112 Gg S).

Despite the good agreement of SOCOL-AERv1 with stratospheric aerosol observations, several aspects of the tropospheric sulfur cycle are treated in a coarse manner. For example, the wet and dry deposition schemes are not interactive, i.e. wet removal of precursor gases and aerosol does not depend on the grid cell precipitation and dry deposition does not depend on the land surface type or particle size. The Model Intercomparison Project on the climatic response to volcanic forcing (VolMIP),
which investigated the response of four chemistry-climate models (CCMs) to the 1815 eruption of Mt. Tambora in Indonesia, highlighted several concerns with the deposition fluxes simulated by SOCOL-AERv1. Compared to the other models with interactive deposition schemes, SOCOL-AERv1 displayed lower sulfate deposition in the midlatitude storm tracks, since its wet deposition scheme is not linked with precipitation. As well, SOCOL-AERv1 overestimated sulfate deposition to polar ice sheets in both the preindustrial background and Tambora cases (Marshall et al., 2018). Improvements to the deposition schemes
in SOCOL-AER are expected to lead to better reconstructions of past volcanic activity from deposited sulfate.

Since Sheng et al. (2015) was published, there have also been substantial updates and changes to the stratospheric aerosol observations. OPC measurements have undergone revision due to a correction in the counting efficiency of the instrument (Kovilakam and Deshler, 2015; Deshler et al., 2019). Updated extinction values are available through the Global Space-based Stratospheric Aerosol Climatology (GloSSAC) project (Thomason et al., 2018). This dataset was used to construct the SAGE-
3$\lambda$ dataset of stratospheric aerosol burdens, which is an input for models in phase 6 of the Coupled Model Intercomparison Project (CMIP6). New in situ measurements of $SO_2$ in the upper troposphere have raised a discussion about the magnitude of the cross-tropopause $SO_2$ flux (Rollins et al., 2017, 2018). Here, we use these newly available datasets to evaluate results from the updated SOCOL-AERv2 model.

This paper outlines the changes that have been made from SOCOL-AERv1 to v2 through the implementation of the new
interactive deposition scheme and other improvements. Section 2 summarizes the changes to the SOCOL-AER code and details the experimental setup of the simulations. Section 3 discusses the results of three types of simulations: year 2000 time-slice runs (Sections 3.1–3.3), 2000–2010 transient runs (Section 3.4), and post-Pinatubo transient runs (Section 3.5). We compare the model with stratospheric aerosol observations, from both non-volcanic background and post-Pinatubo periods, as well as with surface measurements of wet and dry deposition fluxes. Section 3.6 presents updated estimates for the year 2000 atmospheric
sulfur budget. Section 4 draws the conclusions of this work.





## 2    Description of model simulations

### 2.1    Year 2000 time-slice simulations

The development of SOCOL-AERv2 consisted of corrections to the SOCOL-AERv1 code and implementations of new schemes. To compare directly with the reference simulation from Sheng et al. (2015) we run time-slice simulations for the year 2000

at each stage of the code changes. For each model run we simulate 10 years, taking the first 5 years as a spin-up from the initial conditions and analyzing only the last 5 years. In this section, we describe changes in the code for each of the time-slice simulations (summarized in Table 1).

### 2.1.1    Rerunning v1 in T31 (simulation: SHENG31) and T42 (SOCOL-AERv1)

A full description of SOCOL-AERv1 can be found in Sheng et al. (2015), so here we will only summarize the main aspects

of the model. The model originated from the SOCOLv3 chemistry-climate model (Stenke et al., 2013), which consists of the middle atmosphere version of the global circulation model (GCM) European Centre/Hamburg 5 (MA-ECHAM5) and the chemistry model MEZON (Rozanov et al., 1999; Egorova et al., 2003). SOCOLv3 includes 39 hybrid vertical levels ranging from the Earth surface up to 0.01 hPa (80 km). The model is an atmosphere-only model, prescribing global sea surface temperatures and sea ice coverage with observed data from the Hadley Centre (Rayner et al., 2003). The quasi-biennial

oscillation is produced in the model by relaxing the simulated zonal winds in the equatorial stratosphere to observed wind profiles (Stenke et al., 2013).

The chemistry module in SOCOL-AERv1 includes a comprehensive range of stratospheric chemical reactions and a simplified set of tropospheric reactions: of the atmospheric hydrocarbons, only methane photochemistry is included. Sheng et al. (2015) introduced online sulfur chemistry and sulfate aerosol microphysics to the SOCOL model, based on the two-dimensional

sulfate aerosol model AER (Weisenstein et al., 1997). The model considers 8 gaseous sulfur species: OCS, $CS_2$, $H_2S$, DMS, methanesulfonic acid (MSA), $SO_2$, sulfur trioxide ($SO_3$), and $H_2SO_4$. Sulfate aerosol particles are resolved in 40 size bins, ranging in radius from 0.39 nm to 3.2 μm, with sequential bins doubling in volume. Chemical reaction rate coefficients and absorption cross-sections of all reactions, including sulfur reactions, follow the recommendations of Sander et al. (2011). In the aqueous phase, sulfur is described as S(IV) and S(VI) without further speciation. Aqueous oxidation of S(IV) by ozone

($O_3$) and hydrogen peroxide ($H_2O_2$) is calculated by the model using the scheme by Jacob (1986). The aqueous production flux of S(VI) is added to the atmospheric sulfate aerosol tracers, with the flux to each bin proportional to the bin volume.

The microphysical scheme of SOCOL-AERv1 considers the nucleation (Vehkamäki et al., 2002), composition (Tabazadeh et al., 1997), growth through $H_2SO_4$ condensation, evaporation (Ayers et al., 1980; Kulmala and Laaksonen, 1990), coagulation (Fuchs, 1964; Jacobson and Seinfeld, 2004), and sedimentation of sulfate aerosol (Kasten, 1968; Walcek, 2000). SOCOL-

AERv1 employs a crude altitude-varying lifetime approach for tropospheric wet removal of sulfur species, with $H_2SO_4$ and MSA having a mean column lifetime of 5 days, and $SO_2$ having a mean lifetime of 2.5 days (Weisenstein et al., 1997). Dry deposition of $SO_2$, MSA, $H_2SO_4$, and sulfate aerosol is modelled assuming a deposition velocity of 1 cm s$^{-1}$ at the ground. The model is run with operator splitting, so that dynamical quantities are recalculated every 15 minutes, whereas the





chemistry, aerosol microphysics, and radiation schemes are called every 2 hours. Twenty sub-time steps are used for the aerosol microphysical scheme, yielding an aerosol microphysical time step of 6 minutes.

The model's boundary conditions that we use for the year 2000 time-slice simulations are identical to Sheng et al. (2015). $SO_2$ is emitted from anthropogenic and biomass burning sources according to a gridded emission inventory for the year 2000

(Lamarque et al., 2010; Smith et al., 2011) and from continuous volcanic degassing (Andres and Kasgnoc, 1998; Dentener et al., 2006b). DMS fluxes are calculated online using a wind-driven parametrization (Nightingale et al., 2000) and a climatology of sea surface DMS concentrations (Kettle et al., 1999; Kettle and Andreae, 2000). 1 Tg S yr$^{-1}$ $CS_2$ is emitted between the latitudes of 52° S and 52° N (Weisenstein et al., 1997). The mixing ratios of $H_2S$ and OCS are fixed at the surface to 30 pptv (Weisenstein et al., 1997) and 500 pptv (Chin and Davis, 1995; Kettle et al., 2002; Montzka et al., 2007; Commane et al.,

2013), respectively.

To ensure comparability of results with the new development runs for this paper, we have rerun the source code from Sheng et al. (2015) in two experiments. As opposed to the simulations in Sheng et al. (2015), we output the sulfur cycle burdens and fluxes as accumulated rather than instantaneous quantities, to reduce the influence of diurnal cycling on the 12-hourly output of the model. To test the effect of horizontal resolution on the atmospheric sulfur cycle, we ran one simulation at T31 resolution

(~3.75° x 3.75° in latitude/longitude, SHENG31) and one simulation at T42 resolution (SOCOL-AERv1).

### 2.1.2 Dry radius binning scheme (DRYRAD)

In SOCOL-AERv1, the sulfate aerosol is resolved in wet radius bins. Uptake and evaporation of $H_2O$ during transport-induced changes in relative humidity and temperature cause shifts in the sulfate mass distribution, with respect to wet aerosol radius, the coordinate variable. In SOCOL-AERv1, this process was treated by rescaling the number of sulfate aerosol particles, so

that each bin would have the correct $H_2SO_4$ weight percent. Although this procedure guarantees the conservation of the total number of $H_2SO_4$ molecules, it does not conserve the aerosol number density. To amend this, SOCOL-AERv2 resolves the sulfate aerosol distribution in dry radius bins, similar to the approach of other sectional models (e.g., Kleinschmitt et al., 2017). Dry radius bins can also be described as aerosol $H_2SO_4$ mass bins.

The new dry radius bins range from 0.39 nm to 3.2 μm, corresponding to 2.8 to $1.6 \times 10^{12}$ molecules $H_2SO_4$ per particle

(assuming an $H_2SO_4$ density of 1.8 g cm$^{-3}$), with molecule number doubling between bins. Since aerosol microphysics schemes and heterogeneous chemistry on sulfate aerosol require wet aerosol volume and $H_2SO_4$ weight percent, we calculate these quantities for each bin online, based on the grid cell temperature and relative humidity. This change in the dimension variable of SOCOL-AER necessitated several changes to the sulfate condensation and coagulation schemes, to ensure that the transfer of aerosol from bin to bin is based on molecular fluxes rather than aerosol volume fluxes. For calculation of aerosol

radiative properties, a new look-up table was produced as a function of relative humidity and temperature. To ease interpretation of the output, the outputted aerosol bins of SOCOL-AER are re-binned to the previous wet volume binning approach.





### 2.1.3 Mass conservation issues (CONSERVE)

In the CONSERVE simulation, corrections were made to the aerosol microphysics scheme in SOCOL-AER, mainly to improve aerosol mass conservation. In the scheme calculating $H_2SO_4$ condensation and evaporation, the equation for the "effective" mean free path of $H_2SO_4$ molecules in air was corrected to agree with Equation 6 from Hamill et al. (1977). An additional

constraint was added in the $H_2SO_4$ condensation scheme that the flux of $H_2SO_4$ from the gas phase must equal the flux of $H_2SO_4$ into the particle phase. This improves mass conservation in cases when $H_2SO_4$ is depleted below the saturation vapor pressure within one time step. Furthermore, the aerosol sedimentation scheme in SOCOL-AERv1 was not applied within boundary layer levels and sedimenting aerosol from the model level above the boundary layer was artificially removed. In the CONSERVE simulation this is amended: sedimenting aerosol from the model level above the boundary layer is added to the

layer below. Sheng et al. (2015) had implemented several forced mass conservation checks on the total (gas phase and aerosol) $H_2SO_4$ burden in each grid cell. If the total burden had changed by more than 0.1% during aerosol microphysics, $H_2SO_4$ aerosol and gas phase mixing ratios would be scaled to agree with the total $H_2SO_4$ burden before microphysics calculations. These forced mass corrections were found to be unnecessary after the above improvements to the microphysics scheme, and therefore they were removed from SOCOL-AERv2.

### 2.1.4 Merging CCMI additions with SOCOL-AER (CCMI)

Since the publication of Sheng et al. (2015), improvements have been made to the SOCOL model in preparation of the coordinated simulations within the Chemistry Climate Model Initiative (CCMI), mainly related to the improvement of tropospheric chemistry processes (Revell et al., 2015, 2018). Many of these improvements have been merged into SOCOL-AERv2 and have upgraded the representation of chemistry in our model, in particular in the troposphere.

In SOCOL-AERv1, as well as SOCOLv3 (Stenke et al., 2013), ozone-depleting substances (ODS) were transported in three families (short-lived Cl, long-lived Cl, and Br) to save computational cost. With modern supercomputers this treatment is no longer necessary and the ODS species are transported individually. The individual treatment of ODS species avoids a repartitioning of the family members, based on simplified age of air estimates, after each transport step. The chemistry scheme was expanded in the CCMI simulation, as described in Revell et al. (2015). We included the Mainz Isoprene Mechanism (MIM-

1) in SOCOL-AER. This scheme considers the degradation of isoprene and necessitates the addition of 14 organic species and 44 chemical reactions to SOCOL-AER (Pöschl et al., 2000). Additional CO emissions were added to the model to account for the effect of oxidation of non-methane volatile organic compounds (NMVOC) emitted from anthropogenic, biogenic, and biomass burning sources. Lightning NOx is now calculated interactively based on cloud top height (Price and Rind, 1992) and grid cell scaling factors from satellite observations (Christian et al., 2003). A cloud modification factor approach (Chang

et al., 1987) was implemented to account for the effect of clouds on photolysis rates. We derived a new look-up table of photolysis rates averaged over two solar cycles (22 years) from a comprehensive reconstruction of total and spectral solar irradiance (NRLSSI) by Lean et al. (2005), which was used in the CCMI REF-C1 experiment. Additional heating through Hartley and Huggins bands of ozone has also been implemented into SOCOL-AER. As documented by Revell et al. (2018),





N$_2$O$_5$ hydrolysis on tropospheric aerosols is now included in SOCOL-AER. Methanesulfonic acid (MSA) chemistry is solved in the explicit scheme instead of the implicit Newton-Raphson scheme, since otherwise the chemical solver does not properly converge. Reaction rates have been updated or added from the NASA JPL Evaluation no. 18 (Burkholder et al., 2015).

### 2.1.5 Treatment of the boundary layer (BNDLAYER)

5 In the previous simulations, to allow for rapid boundary layer mixing, emissions of chemical species were immediately dispersed over the four lowermost model levels ($\sim$1 km altitude), species with prescribed mixing ratios (including OCS and H$_2$S) were dispersed over the six lowermost levels ($\sim$2 km altitude), and dry deposition of species occurred out of the four lowermost model levels. While such a coarse approach was sufficient for stratospheric applications, it is inadequate for deposition flux and tropospheric lifetime calculations. Instead of emitting the species in multiple lower layers, SOCOL-AERv2 emits only in 10 the first model layer ($\sim$70 m), from which the species are mixed via the model's boundary layer parametrizations. The BND-LAYER simulation tests specifically the effect of using only one model layer for emission and prepares the model for revised dry deposition boundary conditions.

### 2.1.6 Interactive dry deposition (DRYDEP)

We implemented the interactive dry deposition scheme described in Revell et al. (2018) in SOCOL-AERv2, replacing the 15 simple prescribed constant deposition velocities of SOCOL-AERv1. The new treatment is based on the DRYDEP scheme in the EMAC model (Kerkweg et al., 2006, 2009). Dry deposition velocities are calculated using an interactive resistance-based approach, which considers surface properties, the solubility and reactivity of each gas tracer, and the radius and density of aerosol tracers (Wesely, 1989). Effective Henry's law constants for near-neutral pH and reactivity of gas tracers are taken from Wesely (1989). These improvements are tested in the DRYDEP simulation.

### 20 2.1.7 Interactive wet deposition (WETDEP)

An interactive wet deposition scheme was added to SOCOL-AER, based on the SCAV submodule in the EMAC model (Tost et al., 2006). Grid scale variables from ECHAM5 such as liquid and ice water content, cloud cover, convective and large-scale rain and ice formation and precipitation fluxes, and the convective upward mass flux are used by the wet deposition scheme. Since our model does not include a comprehensive cloud aqueous chemistry mechanism, we implemented the EASY2 version 25 of the SCAV submodule (Tost et al., 2007), with a constant pH of 5 for cloud water and rain water. The constant pH of 5 is within the wide range of pH values (3.6–7) measured by several hill cap cloud field campaigns (Sellegri et al., 2003; Marinoni et al., 2004; van Pinxteren et al., 2016). Scavenging coefficients for gas phase species are calculated based on Henry's law equilibrium constants. Scavenging of aerosol is based on a radius-dependent calculation of nucleation and impaction scavenging. During cloud evaporation, all scavenged gas phase species are released to the atmosphere in their original species, whereas evaporating 30 scavenged sulfate aerosol species are transferred to the largest aerosol size bin. The wet deposition scheme is applied to SO$_2$, gaseous H$_2$SO$_4$, and sulfate aerosol, as well as other gas chemical tracers such as O$_3$, HNO$_3$, N$_2$O$_5$, H$_2$O$_2$, etc. The WETDEP





simulation includes this new interactive wet deposition scheme instead of the fixed wet deposition lifetimes used in SOCOL-AERv1.

### 2.1.8 Improvement of aqueous phase chemistry (AQCHEM)

In the SO$_2$ aqueous chemistry subroutine of SOCOL-AERv1, the pH of clouds is prescribed vertically according to Walcek and
Taylor (1986), so that pH equals 3 from the surface to 600 hPa, and 4.5 above 600 hPa. However, this paper reported modelled pH within a single cumulus cloud, where liquid water content increased with height. This pH distribution is therefore not applicable to the whole atmosphere. Although in the interactive wet deposition scheme a constant cloud pH of 5 is used (Section 2.1.7), aqueous phase sulfur chemistry is more sensitive to the choice of pH and therefore a more detailed pH distribution was applied. We use an approximation of the modelled cloud pH from Tost et al. (2007) for the revised aqueous chemistry routine.
Between the surface and 600 hPa, north of 20° N a cloud pH of 5.2 is used and south of 20° N a cloud pH of 4.2 is used. Above the 600 hPa level, a uniform pH of 3.5 is used.

In SOCOL-AERv1, the SO$_2$ oxidized in the aqueous cloud phase is released as aerosol. With the new interactive wet deposition scheme, it is possible to transfer the oxidized SO$_2$ directly to the scavenged aerosol flux in cloud water. The wet deposition routine is called at each dynamical time step (15 minutes), while the aqueous phase chemistry was called at each
chemical time step in SOCOL-AERv1 (2 hours). To transfer the oxidized SO$_2$ flux to the wet deposition scheme directly, it was both logical and technically simpler to synchronize these two processes. In the AQCHEM simulation, the above changes were added to SOCOL-AER and the aqueous phase chemistry is called at each dynamical time step.

### 2.1.9 Final development run for SOCOL-AERv2

Sections 2.1.1–2.1.8 complete the description of improvements in developing SOCOL-AERv2 with one exception, namely how
SO$_2$ oxidation is calculated in clouds in the mixed-phase temperature regime. This is important because the S(IV) to S(VI) conversion only occurs in the liquid phase in the model. Therefore, in the final development simulation (SOCOL-AERv2) we wanted to investigate whether the aqueous SO$_2$ reaction was hampered by the model's representation of the liquid fraction in mixed-phase clouds. It has recently been discussed in the literature that many general circulation models (GCMs) underpredict the supercooled liquid fraction (SLF) observed by satellite products (Komurcu et al., 2014; Cesana et al., 2015; Tan et al.,
2016). The modelled liquid fraction in SOCOL-AERv1 underestimates the fitted CALIOP satellite measurements from Hu et al. (2010) throughout most of the mixed-phase cloud temperature range (Fig. S1). In the SOCOL-AERv2 run, we correct for the influence of the underestimated supercooled liquid fraction on SO$_2$ aqueous chemistry. The ECHAM5 calculated liquid water content (LWC) and ice water content (IWC) are added together and inputted into the aqueous phase chemistry subroutine as total water content (TWC). If the grid cell temperature (T ) is in the mixed phase cloud regime (−38°C to 0 °C), we calculate
the observed supercooled liquid fraction (SLF) from the fitted sigmoid function from Hu et al. (2010):

$$SLF_{Hu} = [1 + \exp(-f(T))]^{-1} \tag{1}$$

$$f(T) = 5.3608 + 0.4025T + 0.08387T^2 + 0.007182T^3 + 2.39 \times 10^{-4}T^4 + 2.87 \times 10^{-6}T^5 \tag{2}$$





where T is the air temperature in °C. The determined SLF can is used to correct LWC in the aqueous chemistry subroutine, i.e. $LWC = SLF_{Hu} \times TWC$.

### 2.1.10 Additional sensitivity runs (ICE-OX, AER-SCAV)

We ran two additional simulations to probe whether the remaining disagreement between observations and SOCOL-AERv2

could be caused by overestimated cross-tropopause fluxes of $SO_2$ and sulfate aerosol. In ICE-OX, the aqueous phase oxidation of $SO_2$ was allowed to occur in ice water as well as liquid water. Increased oxidation of $SO_2$ in the upper troposphere reduces its cross-tropopause flux. In AER-SCAV, the scavenging coefficient of aerosol particles on ice clouds was increased by a factor of 20 from 0.05 to 1. This enhances the removal of sulfate aerosol in the upper troposphere.

## 2.2 Year 2000–2010 transient simulations

In order to compare simulated deposition with observations, the model codes from SOCOL-AERv1 and v2 were used to run two sets of transient simulations from 2000 to 2010. Five ensemble members were simulated for both versions of SOCOL-AER, and plotted results show ensemble means and standard deviations. For the transient simulations we made several updates to the boundary conditions used in Sheng et al. (2015). Anthropogenic emissions were taken from the Community Emissions Data Systems (CEDS), which will be used for CMIP6 simulations (Hoesly et al., 2018). Lana et al. (2011) updated the marine

DMS dataset to include three times as many DMS measurements as the previous dataset (Kettle et al., 1999; Kettle and Andreae, 2000) used by Sheng et al. (2015). Transient degassing volcanic $SO_2$ emissions were taken from Diehl et al. (2012). To represent eruptive emissions, we applied a satellite-derived dataset from Carn et al. (2016). The other data sources for the boundary conditions remained the same as in the time-slice simulations, however transient boundary conditions were included rather than applying repeating year 2000 values.

## 2.3 Pinatubo transient simulations

To verify the updated model's performance under volcanically perturbed conditions we have repeated two experiments from Sukhodolov et al. (2018), modelling the Mt. Pinatubo eruption with 7 and 6 Tg S emitted as $SO_2$. As in Sukhodolov et al. (2018), we simulated five ensemble members with sulfur mass released from 14 to 15 June 1991 and spread between 16 to 30 km. We performed two additional runs with SOCOL-AERv1 and SOCOL-AERv2, including the Pinatubo eruption magnitude

of 7 Tg S but with all other sulfur sources switched off, to check the sulfur mass conservation by analyzing the integrated deposition fluxes. To compare with modelled burdens, we used observational estimates from SAGE-4$\lambda$ and SAGE-3$\lambda$ datasets and from the High-Resolution Infrared Radiation Sounder (HIRS) measurements (Baran and Foot, 1994).





## 3 Results and Discussion

### 3.1 Impacts of performed changes in the development of SOCOL-AERv2

In the following section, we discuss the relevant impacts of each stage of code changes on the atmospheric sulfur cycle. Table 2 lists the stratospheric and tropospheric burdens of $SO_2$ and sulfate aerosol and total deposition fluxes for each time-slice
simulation.

#### 3.1.1 Rerunning SOCOL-AERv1 in T31 and T42 resolution

Since the SHENG31 simulation was rerun for this study, several quantities differ slightly in SHENG31 compared to Sheng et al. (2015) (e.g. 114 vs. 109 Gg S of stratospheric sulfate aerosol). The differences in the quantities could caused by the switch in output format from instantaneous 12 hourly values to mean 12 hourly values. However, the changes are minor and
the overall picture for the sulfur cycle remains unchanged. Refining the horizontal resolution to T42 in SOCOL-AERv1 does not result in substantial changes for the tropospheric and stratospheric aerosol burdens.

#### 3.1.2 Dry radius binning (DRYRAD)

The change from wet radius to dry radius binning reduces the tropospheric aerosol burden and increases the stratospheric aerosol burden in DRYRAD (Table 2). The increased stratospheric aerosol burden can be explained by a decrease in the
effective aerosol radius, leading to a longer stratospheric lifetime. The decrease in tropospheric aerosol burden occurs mainly around the northern hemisphere midlatitudes, where anthropogenic emissions of $SO_2$ are high and thus sulfate particles can grow through condensation. In the DRYRAD version of the model the wet particle radius is no longer restricted to 3.2 μm; accounting for the uptake of water the maximum radius can reach well above 10 μm. The possibility of larger particle formation can lead to enhanced sedimentation velocities and therefore reduced aerosol lifetimes.

#### 3.1.3 Mass conservation fixes (CONSERVE)

The corrections in the sedimentation and $H_2SO_4$ condensation schemes do improve the mass conservation of sulfur species. The tropospheric sulfate aerosol burden increases by 4%, mainly due to the correction of the artificial removal of particles sedimenting from the model level above the boundary layer level. This artificial loss due to sedimentation represents a ∼3 Tg S yr$^{-1}$ sink, since the outputted total sulfur deposition increases by this amount from the DRYRAD to the CONSERVE
simulations. Furthermore, in the DRYRAD simulation the total sum of tropospheric aerosol influxes and outfluxes result in an imbalance of 3339 Gg S yr$^{-1}$, which corresponds to about 8% of the source flux of tropospheric sulfate aerosol. In the CONSERVE simulation, this imbalance is reduced to 63 Gg S yr$^{-1}$, i.e. around 0.1% of the aerosol source flux. These improvements to the model provide more confidence to the outputted sulfur cycle fluxes, which will be used to study the sulfur budget in Section 3.6.





### 3.1.4   CCMI chemistry changes (CCMI)

Including the expanded chemistry set and updates from the CCMI version of SOCOLv3 in SOCOL-AER leads to altered distributions of gas phase species in the troposphere. The relevant change for the tropospheric sulfur cycle are increased mixing ratios of $H_2O_2$, causing increased aqueous conversion of $SO_2$ to S(VI). For this reason, the CCMI simulation shows a

19 Gg S lower $SO_2$ burden and a 10 Gg S larger sulfate aerosol burden in the troposphere than the CONSERVE simulation. Larger OH mixing ratios in the upper troposphere/lower stratosphere (UTLS) reduce the $SO_2$ lifetime, causing 5 Gg lower $SO_2$ and 2 Gg higher sulfate aerosol burdens in the stratosphere. These chemical changes also lead to differences in the Pinatubo simulation, to be discussed in Section 3.5.

### 3.1.5   Boundary layer levels (BNDLAYER)

In BNDLAYER the boundary layer conditions are only implemented for the lowest level of the model. The confinement of the boundary layer conditions to one model level reduces the burdens of $SO_2$ and sulfate aerosol in the troposphere and stratosphere. This is a strong effect with reductions of the tropospheric aerosol burden by 40% and the stratospheric aerosol burden by 20% in BNDLAYER. The first cause is the reduction in effective S emissions into the atmosphere, since $H_2S$ is prescribed to 30 ppt in only one model level, instead of six model levels. Assuming steady-state conditions over the 5 year

averaging period, the 8.5 Tg S $yr^{-1}$ decrease in total sulfur deposition (Table 2) corresponds to an 8.5 Tg S $yr^{-1}$ decrease in the sulfur emissions. Another cause for the $SO_2$ and aerosol burden decrease is that $SO_2$ is emitted close to the surface in BNDLAYER, leading to less dispersion of $SO_2$ in the atmosphere and enhanced dry deposition close to emission regions. The shorter $SO_2$ lifetime reduces its atmospheric burden, as well as reducing the conversion of $SO_2$ to $H_2SO_4$ and subsequent sulfate aerosol formation.

The correct treatment of the lowermost model levels remains difficult and is model-dependent. Owing to their coarse resolution, CCMs cannot resolve the transport of chemical species by rapid boundary layer convection and turbulence. This leaves the boundary layer parametrizations in SOCOL-AER imperfect and the number of model levels that should be included in the emission boundary conditions uncertain. For the subsequent simulations we use the single-level boundary layer treatment.

### 3.1.6   Interactive dry deposition (DRYDEP)

The implementation of interactively calculated dry deposition velocities, compared to the previously included constant dry deposition velocities, results in much longer dry deposition lifetimes for both $SO_2$ and sulfate aerosol. $SO_2$ dry deposition velocities decrease more drastically over land than over ocean in the DRYDEP simulation (Fig. 1). Over land, the $SO_2$ dry deposition velocity is smaller than 1 cm $s^{-1}$, which was the original value set in SOCOL-AERv1. The only locations where $SO_2$ dry deposition velocities are greater than 1 cm $s^{-1}$ are above certain parts of the ocean, due to the high solubility of $SO_2$ in

waters at near-neutral pH. The dry deposition velocities of $SO_2$ agree well with the distribution simulated by the EMAC model (Kerkweg et al., 2006). The resultant longer $SO_2$ dry deposition lifetime increases the tropospheric $SO_2$ burden, the conversion of $SO_2$ to aerosol, and consequently the tropospheric aerosol burden. In addition, dry deposition velocities of sulfate aerosol



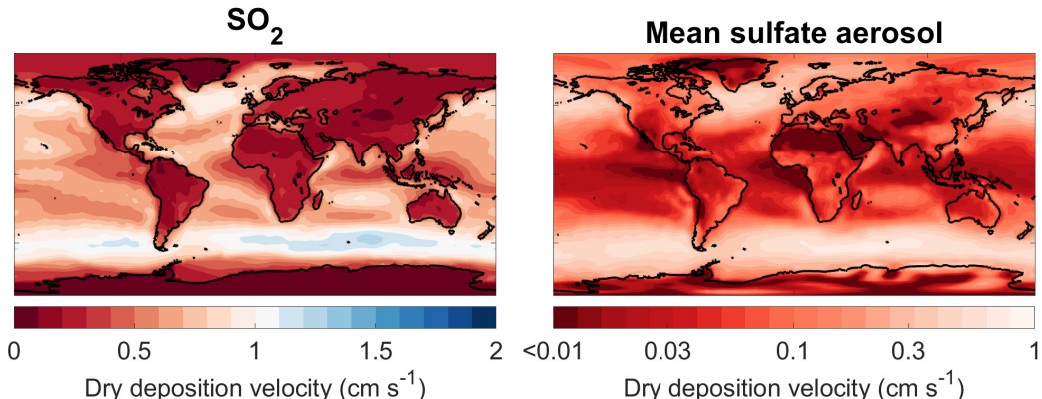

**Figure 1.** Annual mean dry deposition velocities for $SO_2$ (*left*) and sulfate aerosol (*right*) simulated by SOCOL-AERv2. The mean dry deposition velocity for aerosol particles is calculated by weighting the dry deposition velocity for each size bin with the bin's mass concentration at the surface. The color bar highlights differences between the newly simulated deposition velocities and the former homogeneous deposition velocity of 1 cm s$^{-1}$, which is shown in whitish colors on both plots.

decrease globally compared to the assumed constant deposition velocity in SOCOL-AERv1 (1 cm s$^{-1}$), leading to longer aerosol lifetimes with respect to dry deposition. Due to augmented transport of tropospheric $SO_2$ and primary sulfate aerosol from the troposphere, the stratospheric aerosol burden increases by 22 Gg S to 128 Gg S. The changes in DRYDEP largely compensate the changes in BNDLAYER, for which emissions were confined to a single model level.

### 3.1.7 Interactive wet deposition (WETDEP)

When the constant wet deposition lifetimes for sulfur species are replaced with interactively calculated wet removal in the WETDEP simulation, the $SO_2$ wet deposition flux is reduced from 20.1 Tg yr$^{-1}$ to 0.3 Tg yr$^{-1}$, revealing an overestimation in the approach of SOCOL-AERv1. With the elimination of the wet deposition sink for $SO_2$, the tropospheric $SO_2$ burden increases by around 60% and the total (aqueous + gas phase) conversion flux of $SO_2$ to S(VI) increases by around 40%. In Sheng et al. (2015), the mean wet deposition lifetime for $SO_2$ was selected as 2.5 days following the two-dimensional AER model. However, the AER model includes the 2.5 day lifetime for $SO_2$ to account for aqueous oxidation of $SO_2$, which is not explicitly modelled by AER (Weisenstein et al., 1997). As SOCOL-AERv1 already includes a mechanism for aqueous oxidation of $SO_2$ by $H_2O_2$ and $O_3$ in clouds, this resulted in double counting the loss of $SO_2$ by aqueous oxidation in previous simulations. The aerosol wet deposition maps and the relative difference between the DRYDEP and WETDEP simulations are shown in Fig. 2. The inclusion of an interactive wet deposition enhances sulfate aerosol deposition in areas with high precipitation and suppresses it in drier regions. Sulfate deposition is reduced over polar regions, the eastern part of ocean basins, and the Sahara (lower precipitation regions), and is enhanced in the tropics and midlatitude storm tracks (higher precipitation regions). The reductions in sulfate deposition fluxes above Greenland and Antarctica are notable, since SOCOL-AERv1 overestimated the



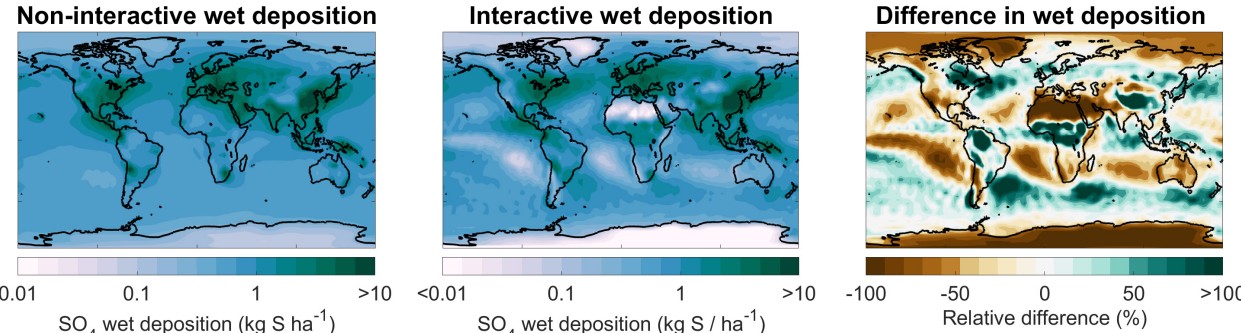

**Figure 2.** Maps of sulfate wet deposition simulated by SOCOL-AER for the year 2000. Three plots are shown: wet deposition in the DRYDEP simulation, which uses the old wet deposition scheme (*left*), wet deposition in the WETDEP simulation, which uses the interactive wet deposition scheme (*center*), relative percent differences in deposition: WETDEP minus DRYDEP (*right*).

magnitude of polar sulfate deposition fluxes compared to ice core measurements, which are used as proxies for past volcanic eruptions (Marshall et al., 2018). Calculating a global aerosol wet deposition lifetime with respect to wet deposition (lifetime = tropospheric aerosol burden divided by aerosol wet deposition flux), DRYDEP has an aerosol wet deposition lifetime of 4.9 days and WETDEP has a lifetime of 5.1 days. Therefore, there is not a large change in the global aerosol wet deposition
lifetime, however the spatial distribution of the wet deposition sink has shifted. In WETDEP the tropospheric column wet deposition lifetime of sulfate aerosol varies from 2 days in the Northern midtlatitude storm tracks to more than 3 years over the southwestern United States. The introduction of interactive wet deposition to SOCOL-AER has the largest impact of any step on the stratospheric sulfate burden. Driven by the longer $SO_2$ wet deposition lifetime, the stratospheric sulfate aerosol burden climbs by around 60% to 202 Gg S. As will be discussed in Section 3.2.1, this value is much higher than the inferred
stratospheric burden from SAGE II data for background non-volcanic conditions. To improve the agreement with observations, we focus on a possible underestimation of the sulfate aqueous chemistry flux, since the unintended double counting of this flux led to good agreement of SOCOL-AERv1 with stratospheric observations (Sheng et al., 2015).

### 3.1.8  Aqueous chemistry changes (AQCHEM)

In the AQCHEM simulation we amended the cloud pH distribution for aqueous chemistry, reduced the aqueous chemistry
time step to 15 minutes, and directly transferred oxidized S(IV) to the scavenged sulfate aerosol in the wet deposition scheme. This increases the aqueous oxidation flux of S(IV) to S(VI) by around 50%. The enhanced aqueous conversion of $SO_2$ to sulfate aerosol leads to increased aerosol formation in the lowermost troposphere, where deposition is efficient. This results in smaller tropospheric burdens of both $SO_2$ and sulfate aerosol, meaning that there is also less transport of tropospheric S to stratosphere. The stratospheric aerosol burden decreases by 18% from 202 Gg S to 165 Gg S. From separate sensitivity studies
(not shown) we find that the shorter aqueous chemistry time step is the main cause of the increased aqueous flux. Goto et al. (2011) investigated the sensitivity of sulfate aqueous chemistry to different settings, and also found that reducing the aqueous





chemistry time step increases the conversion of S(IV) to S(VI). This is because the Henry's law equilibration rate and aqueous oxidation is fast, so with shorter time steps more $SO_2$ can be dissolved in cloud droplets and converted to S(VI).

### 3.1.9 SOCOL-AERv2 and aqueous chemistry in the supercooled liquid fraction

Because of the underprediction of the SLF (Fig. S1) and oxidation of $SO_2$ occuring only in liquid water in SOCOL-AER, the
oxidation of $SO_2$ is likely underestimated in the upper troposphere, leading to too intensive transport of $SO_2$ to the stratosphere. Therefore, in the SOCOL-AERv2 simulation we increased the supercooled liquid fraction to agree with the SLF–temperature relationship observed by CALIOP (Hu et al., 2010). This increase in SLF enhances the $SO_2$ oxidation rate in the middle and upper tropospheric mixed phase clouds, reducing the cross-tropopause $SO_2$ flux by around 10%. However, the impact on the stratospheric aerosol burden is minor, with only a reduction of 6 Gg S ($-4$%) compared to the AQCHEM simulation. Therefore,
the underestimation of the SLF does not play a major role in SOCOL-AER's stratospheric sulfur cycle. However, the amount of $SO_2$ oxidation in the upper troposphere may be affected by other processes, for example by oxidation on ice surfaces. This will be discussed further in Section 3.3.

### 3.2 Comparison of SOCOL-AER versions with stratospheric observations

In this section, we will compare the SOCOL-AERv1 and v2 simulations with observations, to understand how the model results
change in the new version and where deficiencies remain.

#### 3.2.1 Comparison with SAGE-II derived burdens

Sheng et al. (2015) compared the modelled stratospheric sulfate aerosol burden to the value calculated by the SAGE-4$\lambda$ method (Arfeuille et al., 2013). In this method, extinctions measured by the SAGE II satellite product are used to estimate the stratospheric aerosol size distribution, which can then be used to determine the aerosol burden. The background stratospheric
aerosol burden derived from SAGE-4$\lambda$ almost exactly matched the burden simulated by SOCOL-AERv1. Since that time, a new SAGE II retrieval has been published as part of the GloSSAC database (Thomason et al., 2018). A new method (SAGE-3$\lambda$) has been used to calculate the aerosol size distribution from the GloSSAC database. In this method, the surface area density and mass density of very small particles, which are invisible to the satellite extinction measurements, are added to the lognormal size distributions derived from the GloSSAC data. The stratospheric aerosol burden derived from SAGE-3$\lambda$ for the
volcanically quiescent period 2000–2004 is 165 Gg S. This aerosol burden is about 40% larger than the stratospheric burden calculated from SAGE-4$\lambda$, 117 Gg S (note that this value differs slightly from the value reported in Sheng et al. (2015), 112.5 Gg S, possibly due to different assumptions about the tropopause height). The addition of small aerosol particles derived from OPC measurements contribute 18 Gg S to the SAGE-3$\lambda$. The rest of the increase from SAGE-4$\lambda$ to SAGE-3$\lambda$ can be attributed to the new retrieval methods.

The stratospheric aerosol burden simulated by SOCOL-AERv2, 160 Gg S, agrees well with the SAGE-3$\lambda$-derived burden of 165 Gg S. However, evaluating a model's performance with the stratospheric aerosol burden is not straightforward, since both





SAGE-3$\lambda$ and SAGE-4$\lambda$ are themselves derived products and not direct measurements. The retrieval of size distributions from measured SAGE-II wavelengths is uncertain, as can be seen when comparing the change between SAGE-3$\lambda$ and SAGE-4$\lambda$ stratospheric burdens. In addition, the MERRA climatological tropopause height (Rienecker et al., 2011) was used to calculate the stratospheric burden for the SAGE-3$\lambda$ and SAGE-4$\lambda$ products. However, for SOCOL-AER's burden the WMO-defined
tropopause height from the model was used (WMO, 1957). Differences between the tropopause heights used in different calculations can play a big role in the derived burden, since the majority of the aerosol burden is located in the lower stratosphere. For example, if the tropopause height from SOCOL-AER instead of MERRA is used, the stratospheric burdens derived from SAGE-3$\lambda$ and SAGE-4$\lambda$ are around 7% smaller. For these reasons we will evaluate SOCOL-AER with the extinctions measured directly by SAGE-II of version GloSSACv1.0, in addition to the derived burdens.

### 3.2.2 Comparison with SAGE-II extinctions

Figure 3 shows the comparison of annual mean SOCOL-AERv1 and v2 extinctions with SAGE-II at the equator and 45° N for 525 and 1020 nm. Below 20 km at the equator, SOCOL-AERv2 shows higher extinctions at both wavelengths, which match better with the SAGE-II observations. However, in the lowest 1–3 km of the stratosphere, organics are a non-negligible fraction of the overall aerosol burden (Murphy et al., 2014) and therefore can contribute to the aerosol extinction observed
by SAGE-II. If anything, SOCOL-AER as a sulfate aerosol-only model should underestimate the extinction at these altitudes, although to what extent is unknown. Since SOCOL-AERv2 matches or overestimates the SAGE-II extinctions in the lowermost stratosphere, SOCOL-AERv2 may have too high sulfate aerosol concentrations in the lower stratosphere. Between 20 and 25 km SOCOL-AERv2 overestimates the SAGE-II extinctions, while SOCOL-AERv1 matches observations. Between 25 and 30 km both model versions overestimate the SAGE-II extinctions. The model versions are within observed variability between 30
km and 35 km, however above 35 km they tend to underestimate the extinction, possibly because of meteoritic dust, which is the major contributor to extinction in the upper stratosphere (Neely et al., 2011). The comparison at 45° N is similar to the equatorial comparison, with SOCOL-AERv2 overestimating the observed aerosol extinctions below 20 km and otherwise showing similar behaviour to SOCOL-AERv1.

Since the overestimation of SOCOL-AERv2 in the lower stratosphere originates from the introduction of interactive de-
position schemes, it possibly stems from too fast cross-tropopause transport of primary sulfate aerosol and/or SO$_2$, whereas the better agreement of SOCOL-AERv1 may be fortuitous due to the double counting of the SO$_2$ oxidation flux in the wet-deposition scheme. SOCOL-AERv2 is the version that is more physically consistent in its representation of the tropospheric sulfur cycle. However, several outstanding issues remain in SOCOL-AERv2's representation of sulfate aerosol extinction below 20 km at 45° N, and between 20 – 30 km at the equator.

### 3.2.3 Comparison with OPC size distributions

We also compare the SOCOL-AER simulations with in situ OPC measurements from Laramie, USA and Lauder, New Zealand (Fig. 4). Since the publication of Sheng et al. (2015), the counting efficiencies of OPC channels as a function of radius have undergone important revisions (Kovilakam and Deshler, 2015; Deshler et al., 2019). In Fig. 4, we apply the measured counting



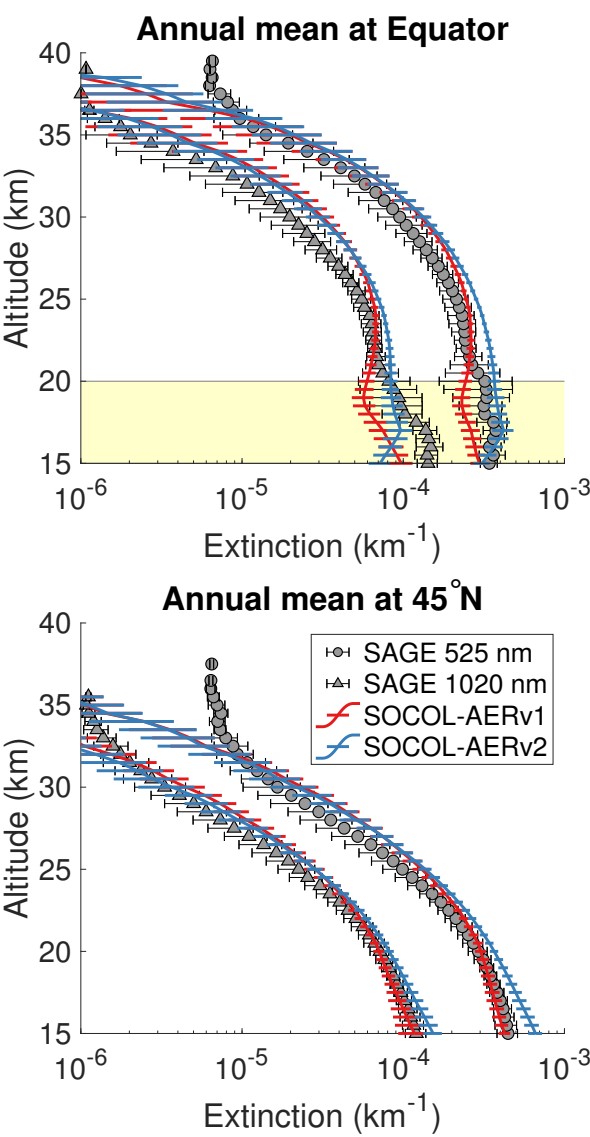

**Figure 3.** Comparison between annual mean model extinctions at 525 and 1020 nm and SAGE II measurements from the GloSSAC project (Thomason et al., 2018) at the Equator (*top*) and 45° N (*bottom*). Observations are averaged between 2000–2004, representing the volcanically quiescent part of the record. Model results are averaged over 5 years of the year 2000 time-slice for SOCOL-AERv1 and SOCOL-AERv2. Horizontal bars represent the modelled or observed standard deviation. The highlighted region in the upper plot corresponds to the altitudes where non-sulfate aerosols may play a role.





efficiencies for the channels r > 0.15 μm, r > 0.25 μm, and r > 0.30 μm from Deshler et al. (2019) to the SOCOL-AER size bins (counting efficiencies were not measured for other channels). In this manner, we can calculate the number density that an OPC instrument would measure given a simulated size distribution.

SOCOL-AERv2 simulates higher number densities of condensation nuclei (CN, r > 0.01 μm) above 25 km, matching the
shape of the observed curve better than SOCOL-AERv1. The transport of polar $H_2SO_4$-rich air to midlatitudes during the breakup of the polar vortex may lead to the high number densities of CN above 25 km in the measurements (Campbell and Deshler, 2014; Sheng et al., 2015). The improved agreement in SOCOL-AERv2 is due to the implementation of dry radius binning and the improvement in sulfur mass conservation, which enable the model to capture the increased transport of CN to midlatitudes during late winter and spring. SOCOL-AERv2 also displays a kink at the tropopause for particle channels
larger than r > 0.15 μm, which appeared after the addition of interactive wet deposition. In the interactive wet deposition scheme, the scavenging efficiency of particles depends on radius, which leads to stronger removal of larger aerosol particles in the troposphere. Lauder OPC measurements may also show a similar kink at the tropopause for the larger particle channels, however it is difficult to verify this given the large variability of the measurements (Fig. 4).

Otherwise, both model versions show similar levels of agreement with the OPC measurements. All four channels larger than
r > 0.25 μm have too high number densities compared to observations at Laramie, with the agreement becoming even worse with altitude. At Lauder the agreement of the largest three channels (r > 0.30 μm) is better. Sheng et al. (2015) attributed the worsening agreement of larger aerosol particles with altitude to either numerical diffusion in the sedimentation scheme or an overestimate in the speed of the Brewer-Dobson circulation, a known artifact in the SOCOL model (Stenke et al., 2013).

### 3.2.4 Comparison with UTLS SO$_2$ measurements

There has been an ongoing debate in the literature regarding the magnitude of the cross-tropopause $SO_2$ flux and its relative importance in establishing the Junge layer (Rollins et al., 2018). The debate has been fueled by a lack of in situ measurements in the UTLS region, and the high temporal and spatial variability of UTLS $SO_2$. Figure 5 compares the tropical UTLS $SO_2$ measured by two aircraft campaigns (Rollins et al., 2017, 2018) with two annual mean satellite products, MIPAS (Höpfner et al., 2015) and ACE-FTS (Doeringer et al., 2012), averaged during volcanically quiescent periods. The two in situ measurements
and the ACE-FTS satellite product all show $SO_2$ mixing ratios of 5 to 10 pptv around the tropical tropopause (∼17 km). MIPAS-observed $SO_2$ is around 24 pptv at 17 km, substantially higher than the other observations.

The annual means of SOCOL-AERv1 and v2 agree with MIPAS-observed $SO_2$ and overestimate the three other observation sets at the tropopause, simulating $SO_2$ mixing ratios between 20 to 30 pptv at 17 km. MIPAS satellite observations of $SO_2$ under non-volcanic conditions are uncertain, which may explain the systematic offset between its $SO_2$ measurements and the
other observation sets (Höpfner et al., 2015; Rollins et al., 2017). On the other hand, in situ measurements lack the spatial and temporal coverage of satellites, which reduces their comparability with global models. More aircraft campaigns will be invaluable to determining the background level of UTLS $SO_2$. If anything, the currently available observations suggest that SOCOL-AER's cross-tropopause $SO_2$ transport might be too high. In Section 3.3 we will investigate the consequences of an overestimated UTLS $SO_2$.

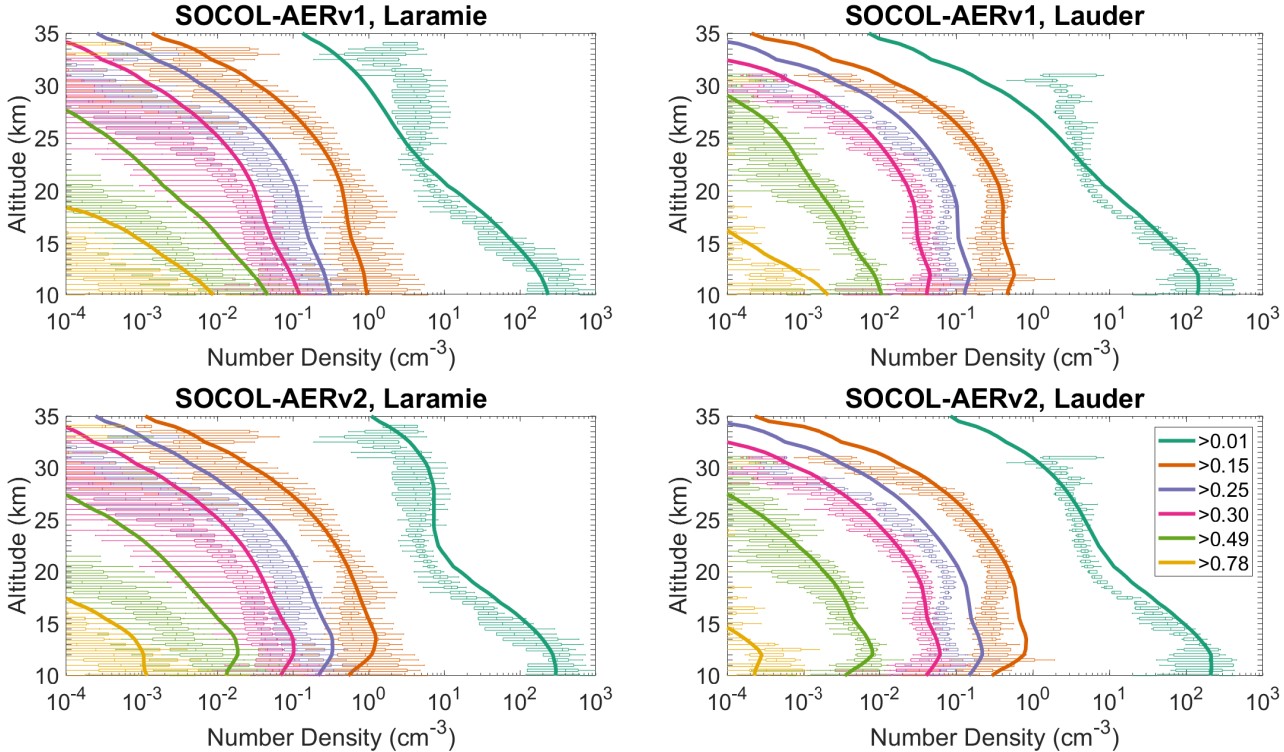

**Figure 4.** Number densities of particle size bins measured by OPC (Deshler et al., 2003; Deshler, 2008) and modelled by SOCOL-AERv1 and v2 over Laramie, Wyoming, USA (41° N, 105° W) and Lauder, New Zealand (45° S, 170° W). Measured number densities are shown as box plots (minimum excluding outliers below the 0.4 percentile, 25th percentile, median, 75th percentile, maximum excluding outliers above the 99.6 percentile) and modelled number densities as solid lines. For the Laramie plots (*left*), OPC measurements are used from the period 1999–2008 and model results are averaged over the 5 years of the time-slice. For the Lauder plots (*right*), OPC measurements are used from January to April 1998–2001 and zonal mean model results are averaged from January to April over 5 years of the time-slice. Model results are weighted with the counting efficiencies for OPC channels from Deshler et al. (2019) for direct comparability with the measurements.

### 3.3 Observational disagreements with SOCOL-AERv2

To summarize, SOCOL-AERv2 shows similar levels of agreement with stratospheric sulfur observations as SOCOL-AERv1. The stratospheric aerosol burden simulated by SOCOL-AERv2, 160 Gg S, agrees very well with the SAGE-3$\lambda$ retrieved burden, 165 Gg S. SOCOL-AERv2 slightly overestimates the SAGE-II aerosol extinction in the lowermost stratosphere at 5 the equator and in the lowermost stratosphere at 45° N, namely by up to 25% at wavelength 1020 nm and by up to 40% at 525 nm. Since organic particles may contribute to the aerosol burden in the lowest 1–3 km of the stratosphere, we think that SOCOL-AERv2 is actually overestimating the cross-tropopause transport of sulfur. OPC measurements also show that large particle channels (r > 0.25 μm) are overestimated in the UTLS at midlatitudes (by up to a factor of 3). A second region where SAGE-II extinctions diverge from the simulated values is between 25 and 30 km at the equator, where SOCOL-AERv2

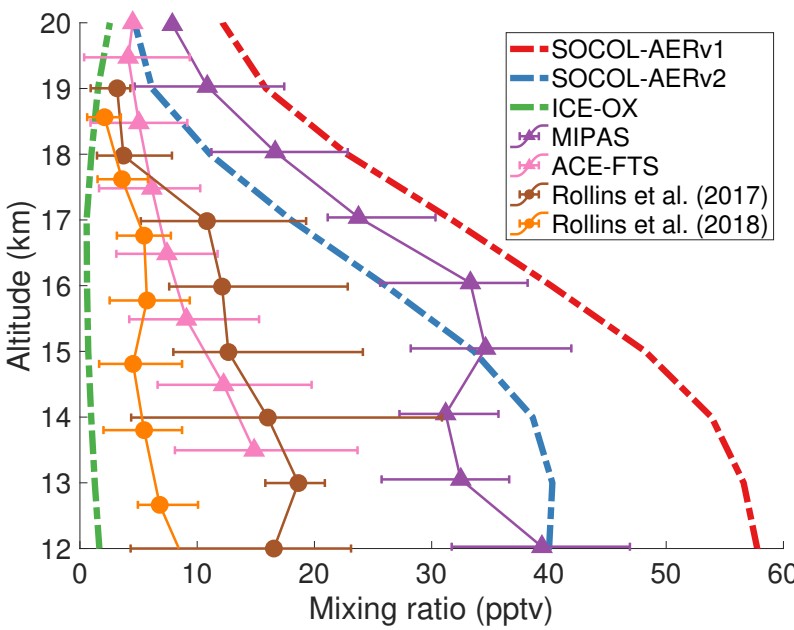

**Figure 5.** Comparing modelled and measured SO$_2$ mixing ratios in the tropical UTLS region between 10–25° N. Modelled results from three simulations are shown as colored dashed lines and are averaged over 5 years of the time-slice. Observational datasets are shown as solid lines with interquartile ranges, extracted from data in Fig. 3 of Rollins et al. (2017) and Fig. 11 of Rollins et al. (2018). The satellite datasets (triangles), MIPAS and ACE-FTS, show mean SO2 values between 2002–2012 and 2004–2010, respectively, and have been filtered to remove any data affected by major volcanic eruptions. Rollins et al. (2017) data (brown circles) represent in situ flight data from October 2015 over the Gulf of Mexico and the tropical eastern Pacific Ocean. Rollins et al. (2018) data (orange circles) were measured in a flight campaign over the tropical western Pacific Ocean in October 2016. Model and satellite data are averaged over all longitudes between 10–25°N.

overestimates extinctions. Above 35 km at the equator and above 30 km at 45° N SOCOL-AERv2 underestimates aerosol extinctions, however this is likely caused by a lack of meteoritic material in the model.

    To address the possible overestimation of sulfur transport to the stratosphere, we ran two additional simulations, AER-SCAV and ICE-OX. In AER-SCAV, we increased the scavenging coefficient of aerosol on ice clouds by a factor of 20, from

5  0.05 to 1, maximizing the effect of upper tropospheric sulfate aerosol removal. In ICE-OX, the ice water content was added to the liquid water content before the aqueous phase chemistry routine, so that SO$_2$ oxidation occurs as well in middle and upper tropospheric ice clouds, maximizing the effect of condensed phase S(IV) to S(VI) oxidation. One laboratory study has identified the SO$_2$ + H$_2$O$_2$ reaction on the surface of ice as a possible sink for SO$_2$, although complicating factors, like partial pressure dependent reaction probabilities and surface poisoning during the reaction, make it difficult to extrapolate

10  the measurements to atmospheric conditions (Clegg and Abbatt, 2001). Furthermore, Rotstayn and Lohmann (2002) found improved model agreement with Arctic sulfate measurements when they included SO$_2$ oxidation also in ice water. In addition, physical uptake of SO$_2$ without conversion to S(VI) on ice has been observed in the laboratory (Huthwelker et al., 2001) and




may lead to gravitational settling; uptake of $SO_2$ on ice is not considered in either SOCOL-AERv1 or v2. Assuming that $SO_2$ oxidation occurs in cloud ice water at the same rate as cloud liquid water is likely an upper limit estimate for the scavenging of $SO_2$ on ice.

These extreme simulations both succeed in reducing the cross-tropopause sulfur transport, leading to strongly reduced strato-
spheric aerosol burdens, namely 133 Gg S in AER-SCAV and 92 Gg S in ICE-OX. In these two simulations extinctions at 45° N now match observations in the lowermost stratosphere, while equatorial extinctions underestimate observations, which may be reasonable since organic aerosol particles play a role in this level (Fig. S2). Similarly, the modelled OPC channels are reduced in number density in AER-SCAV and ICE-OX at midlatitudes (Fig. S3). However, ICE-OX now shows too low number densities of CN in the UTLS compared to OPC measurements, suggesting either that too much $SO_2$ is removed or that other
aerosol types contribute to CN at these altitudes. The available $SO_2$ measurements also imply that too much $SO_2$ is removed in ICE-OX, since the simulated $SO_2$ concentration at 17 km ($\sim$1 pptv) is lower than the in situ and ACE-FTS values of 5 to 10 pptv (Fig. 5).

It is important to mention that although the agreement in the UTLS was improved by AER-SCAV and ICE-OX, there may be other reasons behind the too high cross-tropopause transport in SOCOL-AERv2. Convective transport of $SO_2$ and aerosol
to the upper troposphere may be too strong in SOCOL-AERv2, which is a common problem with other GCMs (Allen and Landuyt, 2014). In this case, chemical oxidation of $SO_2$ on ice or increased aerosol scavenging, rather than being a missing feature in itself, would be compensating for the strong convective transport. This could be further investigated by testing the sensitivity of SOCOL-AER's sulfur cycle to different convection schemes, as has been done for other models (Tost et al., 2010). The choice of the convective scheme and the order in which it is called relative to the wet deposition routine could be used
to further tune SOCOL-AERv2 in the UTLS, however a clear challenge is the lack of in situ measurements at these altitudes. Further measurements of these species in the UTLS would be helpful to constrain the importance of $SO_2$ and primary sulfate aerosol in establishing the stratospheric aerosol layer.

### 3.4 Evaluation of SOCOL-AER deposition in transient simulations

In order to evaluate the performance of SOCOL-AER versions in the troposphere, we will compare simulated annual mean
deposition fluxes with the database compiled for the World Meteorological Organization (WMO) assessment of precipitation chemistry and deposition (Vet et al., 2014). The WMO assessment only included regionally representative sites, e.g. excluding measurements within 50 km of industrial or urban areas, which should be comparable to the simulated values in a global model with coarse resolution. The deposition fluxes reported in the WMO assessment were averaged in 3-year periods, 2000–2002 and 2005–2007. The WMO assessment corrected wet deposition measurements for sea salt contributions of sulfate at
sites less than 100 km from coastlines and at all African measurement sites. For our study, we only use the sites whose measurement methodology and temporal data coverage were assessed as "satisfactory" or "conditional" in the WMO database. We interpolated modelled annual mean deposition to the coordinates of the measurements stations. Several previous model intercomparison projects that simulated deposition (Dentener et al., 2006a; Lamarque et al., 2013; Vet et al., 2014; Tan et al., 2018) will be used as benchmarks for the performance of SOCOL-AER compared with observations.



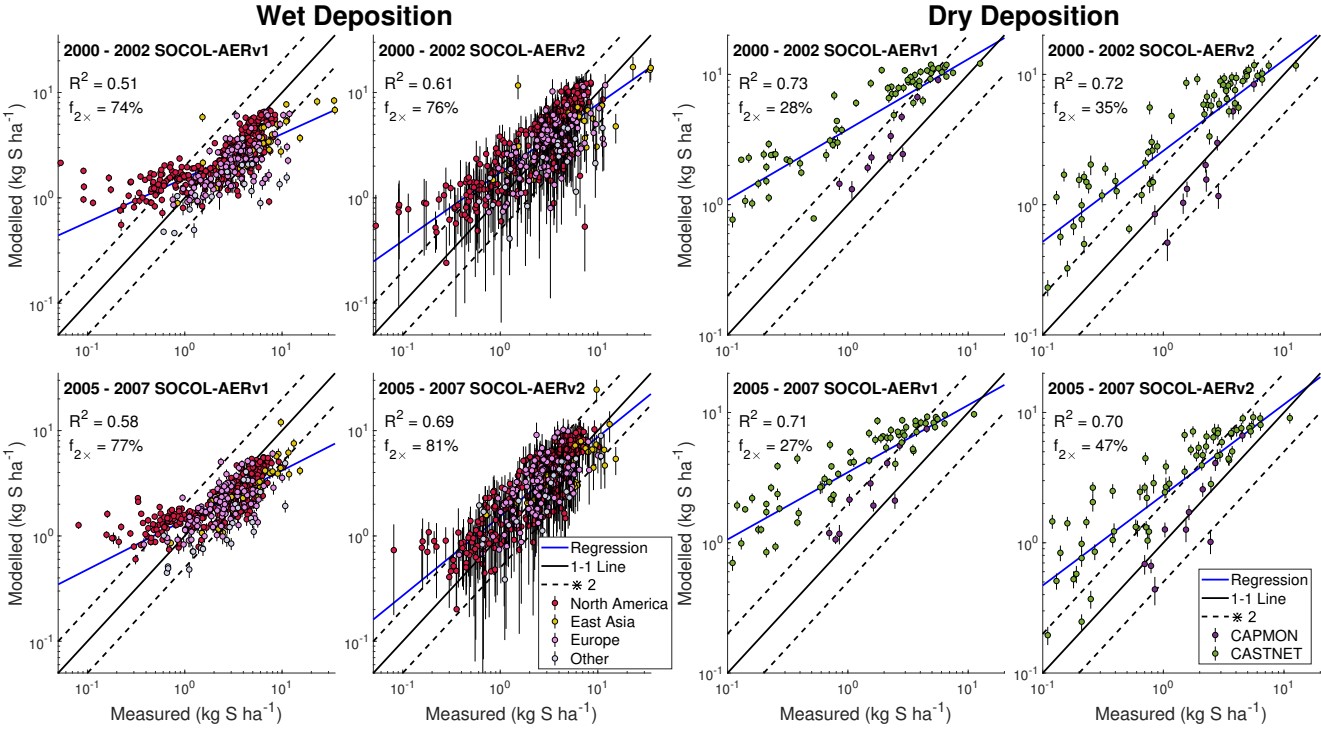

**Figure 6.** Evaluation of modelled total sulfur wet (*left*) and dry (*right*) deposition fluxes against measurement sites from the WMO database (Vet et al., 2014). SOCOL-AERv1 and SOCOL-AERv2 are compared with measurements averaged in two different time periods, 2000–2002 and 2005–2007. The ensemble standard deviation for the model results is shown as vertical bars. A power regression between the simulation results and measurements is shown in blue, and can be compared to the one-to-one line shown in black. Two model evaluation metrics are listed on the plots: the goodness of fit of the power regression between model and measurements ($R^2$) and the fraction of stations where the model is within a factor of 2 of measurements ($f_{2\times}$). Points are colored according to the region (for the wet deposition plots) or the measurement network (for the dry deposition plots) of the measurement stations.

### 3.4.1 Wet deposition

The wet deposition scheme in SOCOL-AERv2 is coupled to the climate model's cloud and precipitation fields, whereas in SOCOL-AERv1 constant wet deposition lifetimes are applied. SOCOL-AERv1 therefore simulates too large deposition fluxes in dry regions and too small deposition fluxes in wet regions compared to SOCOL-AERv2 (Section 3.1.7). To verify the

5  SOCOL-AERv2 wet deposition fluxes in both dry and wet regions, we compare simulated and observed sulfate deposition fluxes over three orders of magnitude in Fig. 6. The presentation is logarithmic, since using linear axes would give too high weight to large deposition fluxes, obscuring biases at the lower range of deposition fluxes.

SOCOL-AERv1 indeed overestimates low deposition fluxes (< 1 kg S ha$^{-1}$), corresponding to sites in drier areas (< 50 cm yr$^{-1}$ precipitation). For both time periods (2000–2002 and 2005–2007), SOCOL-AERv2 improves the agreement with





observations compared to SOCOL-AERv1 ($R^2$ = 0.61 and $R^2$ = 0.69 for SOCOL-AERv2 vs. $R^2$ = 0.51 and $R^2$ = 0.58 for SOCOL-AERv1). The fraction of stations where the model is within a factor of two of observations ($f_{2\times}$) also improves slightly for both measurement periods in SOCOL-AERv2. The variability of simulated wet deposition fluxes, shown by the ensemble standard deviation bars in Fig. 6, increases in SOCOL-AERv2 because wet deposition is coupled to modelled precipitation.

Due to the internal variability of modelled precipitation in a free running climate model, multiple-ensemble simulations as well as long-term deposition measurements are required when comparing models with observations. Overall, SOCOL-AERv2 matches the measurements better than SOCOL-AERv1, especially for sites with low and high deposition fluxes.

Nevertheless, there are remaining biases in the deposition fields of SOCOL-AERv2, for example high biases in many North American sites compared to the WMO observations (Fig. 6 and Table 3). Since the model's deposition scheme is coupled to

precipitation fields, inaccuracies in the modelled precipitation distribution can lead to incorrect deposition fluxes. We calculated the model's precipitation biases compared to the WMO database for each station. The bias in precipitation depth in SOCOL-AERv2 correlates with the bias that we find for sulfate deposition fluxes (Spearman's correlation coefficient, $\rho$ = 0.5). This finding suggests that some of the model biases can be explained by errors in precipitation fields rather than errors in the wet deposition scheme. Both versions of SOCOL-AER match the observations better in the period 2005–2007 rather than 2000–

2002. Since there are no large differences in the precipitation biases between these periods, this improvement could be related to more accurate $SO_2$ emission maps for the 2005–2007 period. One known issue with the CEDS anthropogenic $SO_2$ inventory is that the emissions from western United States are overestimated compared to eastern United States (Hoesly et al., 2018). SOCOL-AERv2 also shows higher deposition biases in the western United States. Since errors in emission inventories and model precipitation fields impact the evaluation of the deposition field, it is difficult to attribute errors to the deposition scheme

itself.

Table 3 compares the performance of SOCOL-AERv2 to past model intercomparison studies, including Photocomp (Dentener et al., 2006a), ACCMIP (Lamarque et al., 2013), HTAP I (Vet et al., 2014), and HTAP II (Tan et al., 2018). These intercomparison projects used observational data from the same networks as in the WMO database to evaluate their results. However, the analysis periods for these intercomparison projects differed from this study for both the simulations and obser-

vations (Table S1), which can contribute to the differences in the results. SOCOL-AERv2 shows similar levels of agreement with observations as the previous model intercomparison studies in the European and East Asian regions. The model biases, correlation coefficients, and fraction of values within ±50% of measurements fall within or very close to the range of the intercomparison projects. In North America, SOCOL-AERv2 correlates similarly with observations (Pearson correlation coefficient, $R$ = 0.8), however biases and the fraction within ±50% are worse than the past intercomparison projects. As mentioned

above, the North American deposition fluxes may be affected by inaccuracies in our model's precipitation fields and/or errors in the anthropogenic $SO_2$ emission inventory. In addition, there are several factors that advantage the model-intercomparison projects compared to the SOCOL-AER runs. Firstly, we have compared ranges of the multi-model means and not the individual model values from the intercomparison projects, which likely have a wider spread and individually worse performance. Secondly, model resolution can play an important role in the comparison between observations and simulations (Tan et al.,

2018), and SOCOL-AER was run at a relatively coarse resolution (2.8° × 2.8°) compared to many of the models used in the





intercomparison project. Finally, three of the intercomparison projects (Photocomp, HTAP I, and HTAP II) were based on off-line chemistry-transport models that are run with observed meteorology. On the other hand, SOCOL-AER is used in free running mode, producing five ensemble members that are subsequently averaged. SOCOL-AER, and its parent climate model ECHAM5, include precipitation biases that impact the simulation of deposition. Considering these aspects, the performance of

SOCOL-AERv2 compares well with state-of-the-art sulfate wet deposition models.

### 3.4.2   Dry deposition

Dry deposition fluxes compiled by the WMO assessment are based on North American stations from the Canadian Air and Precipitation Network (CAPMoN) and the US Clean Air Status and Trends Network (CASTNET). Dry deposition fluxes are not measured directly, but are inferred through surface-based measurements of gas and particle concentrations and estimates

of their dry deposition velocities (Wesely and Hicks, 2000; Vet et al., 2014). The estimation of dry deposition velocities is uncertain; the $SO_2$ dry deposition velocities calculated by the CAPMoN network are around 50% higher than those of the CASTNET network (Schwede et al., 2011). The inferred dry deposition fluxes are therefore less reliable for comparisons with models than the measured wet deposition fluxes.

Fig. 6 displays the comparison between SOCOL-AER versions and the total sulfur (sum of $SO_2$ and aerosol) inferred

dry deposition fluxes from North American measurement networks. Despite the systematic bias between the CAPMoN and CASTNET networks, we calculate the power regression, $f_{2\times}$, and the $R^2$ value based on the combined dataset of both networks. We take this approach as it is unclear which network's dry deposition velocity calculation is more accurate (Wu et al., 2018). SOCOL-AERv2, with its new interactive dry deposition scheme, shows similar correlation with the observations as SOCOL-AERv1. However, SOCOL-AERv2's improved deposition scheme simulates much more realistic slopes of the correlation

lines and higher fractions of model results within a factor of 2 of observations. The improved agreement of SOCOL-AERv2 is likely caused by the better spatial representation of $SO_2$ dry deposition velocities in the interactive scheme. The modelled dry deposition fluxes can also be affected by the new wet deposition scheme, since enhanced sulfur removal through wet deposition leaves less sulfur available for dry deposition, and vice versa. Similar to the wet deposition comparison, the model performs better in 2005–2007 compared to 2000–2002, suggesting that the North American emission inventories may be biased

in the earlier time period. Additional comparisons for the $SO_2$ and aerosol dry deposition fluxes with observations show better agreement for SOCOL-AERv2 than SOCOL-AERv1 (Fig. S4 and S5).

Compared to past model-intercomparison projects for total sulfur dry deposition, SOCOL-AERv2 simulates a lower bias and a higher fraction of model values within ±50% of the observation sites (Table 4). The agreement nevertheless remains poor, with only 19 to 28% of the modelled total dry deposition fluxes within ±50% of the observations. Aerosol dry deposition

biases are larger in SOCOL-AER compared to past model intercomparison projects. However, since $SO_2$ dominates the dry deposition flux (compare Fig. S4 and S5), the reduced biases in the $SO_2$ deposition fluxes in SOCOL-AERv2 leads to overall lower total sulfur dry deposition biases. It is difficult to conclude whether the observations should actually be the target for the model in this case, since dry deposition fluxes are inferred values with inherent uncertainties. As observational networks improve their parameterizations for deriving the dry deposition flux, they will become more reliable standards with which to





compare modelled results. However, the similar, if not better, agreement of SOCOL-AERv2 compared to sulfur dry deposition from model-intercomparison studies adds confidence to the implemented dry deposition scheme.

## 3.5 Pinatubo simulation with SOCOL-AERv2

The eruption of Mt. Pinatubo in 1991 remains the strongest directly observed volcanic event, which makes it a valuable test for models. Sukhodolov et al. (2018) analyzed the performance of SOCOL-AERv1 for this case in a series of sensitivity experiments, demonstrating reasonable agreement with observations of different aerosol parameters. Figure 7 shows the SOCOL-AERv1 and v2 global total stratospheric aerosol burden for emission estimates of 6 and 7 Tg S compared to HIRS, SAGE-$3\lambda$ (v4), and SAGE-$4\lambda$. The same Mt. Pinatubo emission estimates modelled with SOCOL-AERv2 show clear differences compared to SOCOL-AERv1, expressed in both the shape of the peak and its magnitude. SOCOL-AERv2 is $\sim$0.4 Tg S higher at the peak values in late 1991 but lower at the tail after mid-1992. The main reasons for a narrower and stronger peak in SOCOL-AERv2 are the changes to the chemistry scheme for the CCMI simulation and the update of reaction coefficients to Burkholder et al. (2015) recommendations (Section 2.1.4), which led to higher OH concentrations in the UTLS. In SOCOL-AERv2, the oxidation of $SO_2$ is therefore faster, causing faster aerosol formation and its earlier removal from the stratosphere. Another change that contributed to the differences is the improved sulfur mass conservation. This effect is illustrated by the integrated total sulfur deposition in the two model versions, after the emission of 7 Tg S with all other sulfur emissions switched off (orange and grey lines in Fig. 7). This experiment shows that by the end of 1995, when almost all the volcanic aerosol is already removed from the atmosphere, SOCOL-AERv1 deposited only 6.4 Tg S, while in SOCOL-AERv2 this is now improved to 6.7 Tg S, i.e. the volcanic sulfur mass loss decreased from 8.6% to 4.2%. The remaining part of the mass loss is likely due to limitations of the transport scheme (Stenke et al., 2013). Overall, in SOCOL-AERv2 the Mt. Pinatubo emission estimate required for the model to reproduce the observed burden peak is still between 6 and 7 Tg S, given the large observational uncertainty.

## 3.6 Updated non-volcanic sulfur budget for year 2000

Figure 8 shows an update from SOCOL-AERv2 for the atmospheric sulfur budget under volcanically quiescent background conditions. The model was run with repeating boundary conditions for the year 2000, with the updated sulfur emission datasets (Section 2.2). Due to higher emissions in the updated sulfur emission datasets, the stratospheric aerosol burden shown in Fig. 8 (167 Gg S) is slightly larger than the burden for the SOCOL-AERv2 run in Table 2 (160 Gg S). We have added diagnostics to track tracer fluxes within the model's planetary boundary layer (PBL), expanding from the budget figure from SOCOL-AERv1 (Fig. 3 in Sheng et al. (2015)). For these calculations, we extracted the modelled height of the PBL from the ECHAM5 vertical diffusion routines. In Fig. 8, italicized fluxes are calculated from the other outputted fluxes assuming steady-state conditions, i.e. that the fluxes into/from a species add up to zero. The fluxes in the PBL do not fully balance, due to failures in the steady-state assumption, difficulties in extracting chemical fluxes from the iterative chemical solver, or remaining mass conservation errors in the model. However, the imbalances in the PBL are at most 4% of the total input fluxes in the PBL for each species. Therefore, the outputted fluxes shown in Fig. 8 can be considered reliable.

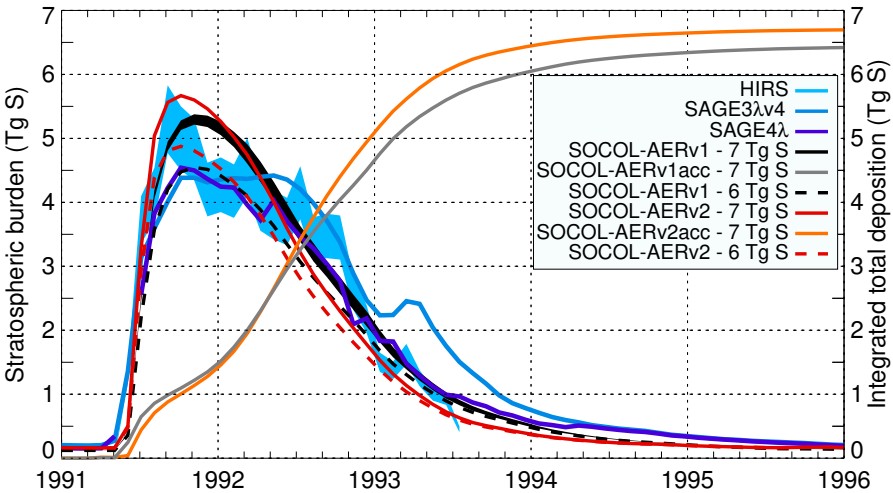

**Figure 7.** Time evolution of the globally averaged stratospheric aerosol burden calculated by SOCOL-AERv1 (Fig. 1a from Sukhodolov et al. (2018)) and SOCOL-AERv2 compared with the HIRS and SAGE II-derived data (SAGE-$4\lambda$ and SAGE-$3\lambda$v4). Light blue shaded area: uncertainties of HIRS. Black shaded area: $2\sigma$ 5-member ensemble spread of one of the model experiments; others are shown as ensemble means. Model experiments are performed with two emission estimates: 6 and 7 Tg S. Two accumulated (acc) lines in orange and grey represent modelled globally integrated deposition to check the mass balance.

    Figure 8 reveals differences between the tropospheric sulfur cycle in the PBL and the free troposphere. OCS, MSA, and sulfate aerosol all show higher burdens in the free troposphere compared to the PBL, since these species are produced chemically in the atmosphere and/or they are long-lived species. On the other hand, several sulfur species ($CS_2$, DMS, $H_2S$, and $SO_2$) have higher burdens in the boundary layer compared to the free troposphere, despite the free troposphere having more

volume. Emission from the surface is the most important atmospheric source for these species. Still, free tropospheric $SO_2$ is supplied not only by cross-PBL transport (67%), but also through transport and oxidation of short-lived species (mainly DMS and $H_2S$). This result supports another modelling study that highlighted the transport pathway of DMS to the upper troposphere by deep convection over the ocean (Marandino et al., 2013). Aqueous phase $SO_2$ oxidation dominates oxidation of $SO_2$ in both parts of the troposphere in SOCOL-AERv2 (74% of total $SO_2$ oxidation in the PBL and 68% in the free troposphere). Sulfate

aerosol is produced in the PBL through condensation of $H_2SO_4$ on existing particles and aqueous phase oxidation of $SO_2$, with negligible nucleation fluxes. In the upper troposphere, nucleation of new $H_2SO_4$–$H_2O$ droplets becomes more important, although growth of existing particles remains the largest mass flux to the particle phase. Figure 8 shows a large downward flux of aerosol from the free troposphere to the PBL (18 Tg S $yr^{-1}$). This balanced flux does not represent only gravitational sedimentation to the PBL, but rather mostly wet removal of free tropospheric aerosol to the surface. The PBL diagnostics in

SOCOL-AERv2 are a useful tool for understanding transport and transformation of sulfur species in the troposphere.



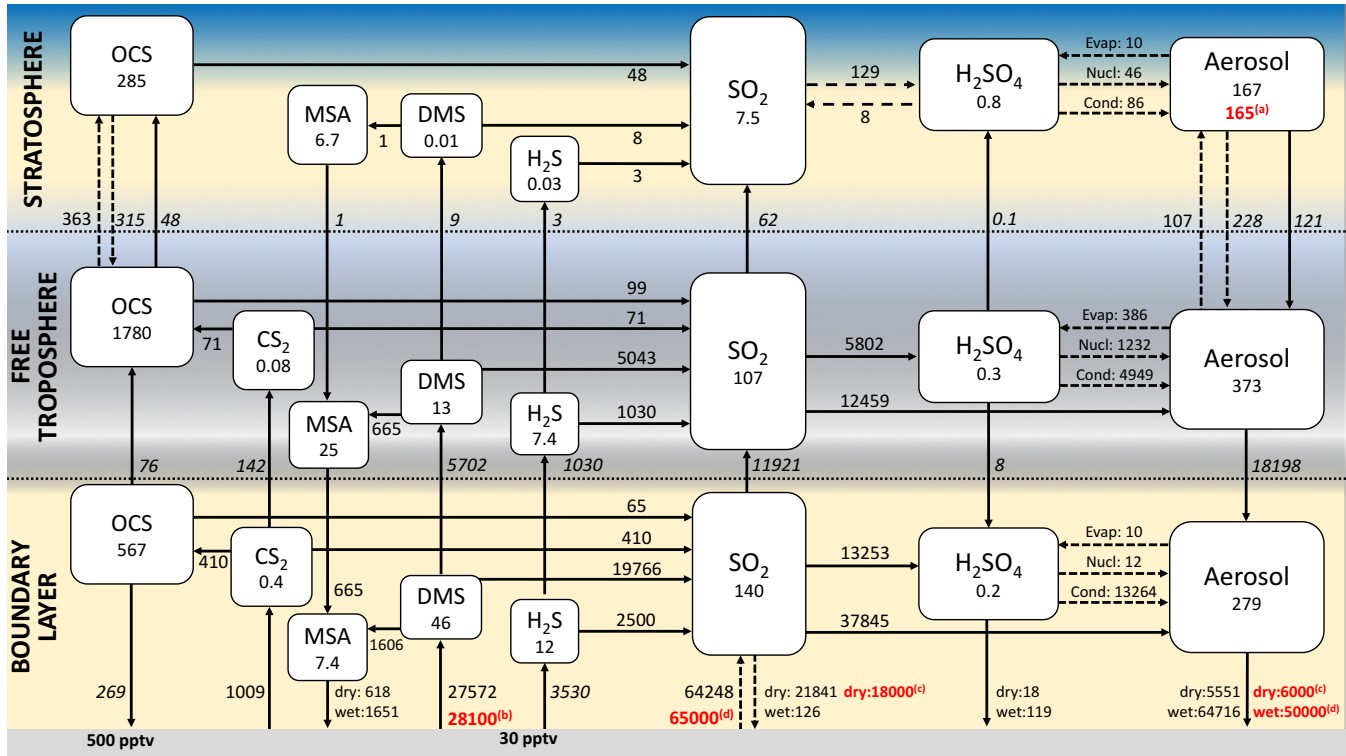

**Figure 8.** Atmospheric sulfur budget from SOCOL-AERv2 under volcanically quiescent conditions for the year 2000. The figure is produced using output diagnostics that track sulfur fluxes and burdens within the planetary boundary layer (PBL), free troposphere, and stratosphere. Solid arrows show net fluxes and dashed arrows show one-way fluxes, all in Gg S yr$^{-1}$. Simulated burdens of sulfur species are given within the boxes, in units of Gg S. Italicized numbers represent fluxes that are derived by balancing other fluxes, assuming steady state of the upper layers. For example, net cross-tropopause fluxes are calculated by balancing the stratospheric chemical fluxes and net cross-PBL fluxes are calculated by balancing the free tropospheric chemical fluxes. Upward OCS and sulfate aerosol cross-tropopause fluxes are calculated based on the residual meridional and vertical air velocities ($v^*$ and $w^*$) and concentrations at the tropopause. Black numbers: SOCOL-AERv2 results. Red numbers: (a) SAGE-3$\lambda$ stratospheric aerosol burden (b) DMS emissions estimated by (Lana et al., 2011) (c) dry deposition fluxes from NCAR-CAM3.5 model (Lamarque et al., 2012) for the year 2000, which participated in ACCMIP (d) multi-model mean wet deposition from ACCMIP models for year 2000 (Lamarque et al., 2013).

SOCOL-AERv1 calculated a total flux to stratospheric aerosol of 181 Gg S yr$^{-1}$, by summing the net cross-tropopause fluxes of gaseous sulfur species and the upward cross-tropopause flux of primary sulfate aerosol (Sheng et al., 2015). With the modifications that were made in this paper, SOCOL-AERv2 now simulates a flux of 228 Gg S yr$^{-1}$ into stratospheric aerosol. Of this flux, 46% is due to upward transport of primary tropospheric aerosol, 27% due to SO$_2$, 21% due to OCS, 4% due to

5  DMS, and 1% due to H$_2$S. These contributions are very similar to the contributions reported for SOCOL-AERv1 (Sheng et al., 2015). As discussed in Section 3.3, the AER-SCAV and ICE-OX runs suggest that too much SO$_2$ and tropospheric aerosol are



transported across the tropopause in SOCOL-AERv2. Future improvements of convection schemes and increased availability of reliable observational data will further constrain the accuracy of the $SO_2$ contribution to the stratospheric aerosol layer.

## 4 Conclusions

For SOCOL-AER to be used to study the tropospheric sulfur cycle, as well as the deposition response to volcanic eruptions,
we implemented new features and applied several corrections to the code. Compared to SOCOL-AERv1, the implemented interactive deposition schemes in SOCOL-AERv2 result in much improved agreement with measurements from sulfur deposition networks. With respect to stratospheric aerosol observations, SOCOL-AERv2 shows similar levels of agreement as SOCOL-AERv1. The modelled estimate for the burden of background stratospheric aerosol has increased from 116 Gg S in SOCOL-AERv1 to 160 Gg S in SOCOL-AERv2. At the same time, the burden derived from SAGE extinctions and Optical
Particle Counter measurements, increased from the SAGE-$4\lambda$ estimate of 117 Gg S by 40% to the most recent estimate from the SAGE-$3\lambda$ dataset of 165 Gg S. Given the uncertainty in the SAGE-$3\lambda$ and SAGE-$4\lambda$ estimates, this might be to some degree fortuitous. Therefore, it is more reliable to compare the model with the satellite extinction measurements directly. Aerosol extinctions in the lower stratosphere are overestimated by SOCOL-AERv2, more than SOCOL-AERv1. We speculate that this might be related to inaccuracies in the model's convective transport scheme, which were also present in SOCOL-AERv1 but
compensated by double counting the $SO_2$ aqueous oxidation flux. Nevertheless, uncertain processes related to $SO_2$ and aerosol scavenging by ice clouds could also lead to overestimation of lower stratospheric sulfate aerosol. Disagreement of $SO_2$ measurements in the UTLS region render improvements of models difficult. Our tests for Pinatubo showed that SOCOL-AERv2 now gives more aerosol mass in 1991 due to faster $SO_2$ oxidation. Better sulfur mass conservation allowed us to decrease the sulfur mass loss after Pinatubo from 8.6% to 4.2%. With the improved mass conservation in SOCOL-AERv2, we are also able
to separate free tropospheric from PBL fluxes in the atmospheric sulfur budget, revealing that short-lived sulfur species (DMS and $H_2S$) contribute strongly to $SO_2$ in the free troposphere.

The model developments presented here increase the applicability of SOCOL-AER to scientific questions in both the troposphere and stratosphere. Namely, due to its improved deposition fluxes, SOCOL-AERv2 is more suitable for: comparison with ice core-derived magnitudes of past volcanic eruptions and their atmospheric impacts (e.g., Marshall et al., 2018); modelling
the atmospheric budget of cosmogenic isotopes, which attach to sulfate aerosols (Delaygue et al., 2015); and studying future changes to sulfur deposition, relevant to agriculture and ecosystems (e.g., Vet et al., 2014). With its updated chemistry and improved sulfur mass conservation compared to SOCOL-AERv1, SOCOL-AERv2 is more reliable for studying the impacts of volcanic eruptions and stratospheric sulfate geoengineering.

*Code and data availability.* Since the SOCOL-AER code is based on ECHAM5, users must first sign the ECHAM5 license agreement
before access to the SOCOL-AER code (http://www.mpimet.mpg.de/en/science/models/license/). SOCOL-AER code is then freely available upon request from the authors. The simulation data presented in this paper are available at: https://doi.org/10.3929/ethz-b-000342078



(Feinberg et al., 2019). SAGE II data from the GloSSAC database can be found online at: https://doi.org/10.5067/GloSSAC-L3-V1.0. The OPC measurements from the University of Wyoming were obtained at ftp://cat.uwyo.edu/pub/permanent/balloon/Aerosol_InSitu_Meas/. Deposition flux measurements can be downloaded online from the World Data Centre for Precipitation Chemistry at http://www.wdcpc.org/global-assessment-data.

*Author contributions.* AF conducted most of the simulations, analyzed the data, and drafted the paper. TS conducted and analyzed the simulations for the post-Pinatubo period. AF was supervised directly by LW, TP, and AS during the model development work. All authors participated in the model development, discussions about the results, and revisions to the manuscript.

*Competing interests.* The authors declare that they have no conflict of interest.

*Acknowledgements.* This work was supported by a grant from ETH Zürich under the project ETH-39 15-2. Thanks to Jianxiong Sheng for
providing plotting codes to analyze SOCOL-AER results and for providing further information about SOCOL-AERv1. Thanks to Debra Weisenstein for discussions about the original AER code. Thanks to Jean-François Lamarque for providing the sulfur deposition data for the NCAR-CAM3.5 model. We acknowledge the researchers behind the GloSSAC SAGE II data and the OPC measurements for use of their data. We also gratefully acknowledge the sources of precipitation chemistry and deposition data acknowledged on page 92 of Vet et al. (2014) Atmospheric Environment, 93, http://dx.doi.org/10.1016/j.atmosenv.2013.10.060. TS and ER acknowledge support from the Swiss National
Science Foundation under grants 200021-169241 (VEC) and 200020_182239 (POLE).





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





**Table 1.** Description of the year 2000 time-slice simulations that were produced during the development of SOCOL-AERv2. Each simulation builds on the previous one, adding new features or correcting a certain process in the model. Two additional sensitivity runs are also listed that reduce the cross-tropopause transport of $SO_2$ and tropospheric sulfate aerosol.

| Simulation name | Description | Section |
|---|---|---|
| SHENG31 | Model version (in T31 resolution) used in Sheng et al. (2015) | 2.1.1 |
| SOCOL-AERv1 | SHENG31 model code, but using T42 resolution | 2.1.1 |
| DRYRAD | Implementation of dry sulfate aerosol radius as the binning scheme, instead of wet radius | 2.1.2 |
| CONSERVE | Improvement of mass conservation in aerosol microphysical schemes | 2.1.3 |
| CCMI | Expansion of tropospheric chemistry scheme (Revell et al., 2015, 2018) | 2.1.4 |
| BNDLAYER | Limiting emission and deposition boundary conditions to the lowermost model layer | 2.1.5 |
| DRYDEP | Addition of interactive dry deposition scheme (Kerkweg et al., 2006, 2009) | 2.1.6 |
| WETDEP | Addition of interactive wet deposition scheme (Tost et al., 2006) | 2.1.7 |
| AQCHEM | Fixes to the aqueous chemistry scheme (time step, transfer to wet deposition flux, cloud pH) | 2.1.8 |
| SOCOL-AERv2 | Correcting the supercooled liquid fraction to CALIOP-observed values for aqueous chemistry | 2.1.9 |
| **Additional runs based on SOCOL-AERv2** | | |
| ICE-OX | Oxidation of S(IV) to S(VI) occurs in cloud ice water, in addition to liquid water | 2.1.10 |
| AER-SCAV | Aerosol scavenging coefficient on ice is increased by a factor of 20 | 2.1.10 |



**Table 2.** Global annual mean sulfate aerosol and $SO_2$ burdens in the troposphere and stratosphere for all volcanically quiescent time-slice (year 2000) simulations in the development of SOCOL–AERv2. Total sulfur deposition (last column) is listed as a check of whether the mass balance of the model has changed. Model results are compared to two datasets, SAGE-3λ and SAGE-4λ, which derived stratospheric aerosol burdens from satellite extinction measurements. The model tropopause is used to separate tropospheric and stratospheric burdens, whereas for the SAGE-3λ and SAGE-4λ calculations the MERRA tropopause is used. The WMO Assessment (Vet et al., 2014) calculates a multi-model mean total sulfur deposition flux of 84.8 Tg yr$^{-1}$ and total emissions of 89.0 Tg yr$^{-1}$ for 2001. Simulated inter-annual variability in one simulation is on the order of 1–2% for burdens, and 0.3% for total deposition.

| Simulation | Tropospheric aerosol burden (Gg S) | Stratospheric aerosol burden (Gg S) | Tropospheric $SO_2$ burden (Gg S) | Stratospheric $SO_2$ burden (Gg S) | Total sulfur deposition (Tg S yr$^{-1}$) |
|---|---|---|---|---|---|
| SHENG31 | 395 | 114 | 259 | 12.4 | 99.8 |
| SOCOL–AERv1 | 397 | 116 | 261 | 11.9 | 99.8 |
| DRYRAD | 367 | 125 | 261 | 12.0 | 97.9 |
| CONSERVE | 382 | 128 | 261 | 12.3 | 101.1 |
| CCMI | 392 | 130 | 242 | 7.1 | 101.0 |
| BNDLAYER | 236 | 106 | 137 | 5.8 | 92.5 |
| DRYDEP | 508 | 128 | 218 | 6.6 | 92.8 |
| WETDEP | 769 | 202 | 351 | 8.2 | 93.6 |
| AQCHEM | 667 | 166 | 245 | 6.6 | 94.5 |
| SOCOL–AERv2 | 640 | 160 | 217 | 6.3 | 94.4 |
| **Additional sensitivity tests** | | | | | |
| ICE-OX | 613 | 92 | 188 | 3.9 | 94.2 |
| AER-SCAV | 579 | 133 | 215 | 6.4 | 94.5 |
| **Observational datasets** | | | | | |
| SAGE-4λ | | 117 | | | |
| SAGE-3λ | | 165 | | | |
| WMO Assessment | | | | | 84.8 |





**Table 3.** Metrics comparing the agreement of simulated sulfate wet deposition with the WMO database, separated by measurement region. Results from SOCOL-AERv2 from two time periods are compared with the range of values from past model intercomparison projects (MIP), including Photocomp (Dentener et al., 2006b), HTAP I (Vet et al., 2014), ACCMIP (Lamarque et al., 2013), and HTAP II (Tan et al., 2018). Note that the observational and simulation time period covered by the other model intercomparison projects differs from SOCOL-AER in this study (see Table S1). The metrics are calculated in linear space, to conform with past MIPs: linear fit slopes differ from the power regression in Fig. 6 and fractions of sites where the model is within ±50% of observations differ from the fraction within a factor of 2 listed in Fig. 6.

| Metric | North America | | | Europe | | | East Asia | | |
|---|---|---|---|---|---|---|---|---|---|
| | Range of MIPs | SOCOL-AERv2 2000–2002 | SOCOL-AERv2 2005–2007 | Range of MIPs | SOCOL-AERv2 2000–2002 | SOCOL-AERv2 2005–2007 | Range of MIPs | SOCOL-AERv2 2000–2002 | SOCOL-AERv2 2005–2007 |
| Mean observations (kg S ha$^{-1}$) | 2.5–3.1 | 3.3 | 3.1 | 2.3–4.0 | 3.8 | 2.9 | 6.5–6.9 | 8.3 | 7.8 |
| Mean model (kg S ha$^{-1}$) | 2.8–3.6 | 4.4 | 4.0 | 2.0–4.6 | 3.7 | 3.5 | 3.9–5.0 | 5.8 | 5.9 |
| Mean bias (kg S ha$^{-1}$) | −0.2–0.5 | 1.1 | 0.9 | −1.3–0.5 | −0.1 | 0.6 | −2.9 – −1.6 | −2.5 | −1.9 |
| Linear fit slope (observations vs. model) | 0.6–1.0 | 1.2 | 1.1 | 0.3–0.6 | 0.5 | 0.6 | 0.3–0.5 | 0.4 | 0.4 |
| Pearson correlation coefficient, $R$ | 0.8–0.9 | 0.8 | 0.8 | 0.6–0.7 | 0.6 | 0.5 | 0.6–0.9 | 0.8 | 0.7 |
| Fraction of sites within ±50% (%) | 70–77 | 48 | 60 | 53–86 | 72 | 61 | 69–88 | 66 | 73 |

**Table 4.** Metrics comparing the agreement of simulated dry deposition with the WMO database in North America, separated by measurement quantity (total sulfur, $SO_2$, and sulfate aerosol). Results from SOCOL-AERv2 from two time periods are compared with the range of values from past model intercomparison projects (MIP), including HTAP I (Vet et al., 2014), ACCMIP (Lamarque et al., 2013), and HTAP II (Tan et al., 2018). Note that the observational and simulation time period covered by the other model intercomparison projects differs from SOCOL-AER in this study (see Table S1). The metrics are calculated in linear space, to conform with past MIPs: linear fit slopes differ from the power regression in Fig. 6 and fractions of sites where the model is within ±50% of observations differ from the fraction within a factor of 2 of observations listed in Fig. 6.

| Metric | Total sulfur dry deposition | | | SO₂ dry deposition | | | Sulfate aerosol dry deposition | | |
|---|---|---|---|---|---|---|---|---|---|
| | Range of MIPs | SOCOL-AERv2 2000–2002 | SOCOL-AERv2 2005–2007 | Range of MIPs | SOCOL-AERv2 2000–2002 | SOCOL-AERv2 2005–2007 | Range of MIPs | SOCOL-AERv2 2000–2002 | SOCOL-AERv2 2005–2007 |
| Mean observations (kg S ha⁻¹) | 1.1–2.3 | 2.5 | 2.0 | 0.8–1.9 | 2.0 | 1.6 | 0.2–0.4 | 0.4 | 0.4 |
| Mean model (kg S ha⁻¹) | 3.6–5.1 | 4.9 | 3.8 | 3.2–4.6 | 3.8 | 2.9 | 0.4–0.5 | 1.1 | 0.9 |
| Mean bias (kg S ha⁻¹) | 2.5–3.7 | 2.4 | 1.7 | 2.4–2.6 | 1.7 | 1.2 | 0.1–0.2 | 0.7 | 0.5 |
| Linear fit slope (observations vs. model) | 1.0–2.7 | 1.3 | 1.1 | 1.0–2.7 | 1.1 | 0.9 | 1.0–1.6 | 2.1 | 2.0 |
| Pearson correlation coefficient, R | 0.8 | 0.8 | 0.8 | 0.8 | 0.8 | 0.8 | 0.8–0.9 | 0.6 | 0.6 |
| Fraction of sites within ±50% (%) | 6–13 | 19 | 28 | 5–6 | 22 | 31 | 47–48 | 19 | 29 |