# Peer review of "Improved tropospheric and stratospheric sulfur cycle in the aerosol-chemistry-climate model SOCOL-AERv2"

_Geoscientific Model Development, 2019_

## Referee Comment (RC1) · Claudia Timmreck (Referee) · 17 Jun 2019

The paper describes the various improvements and development steps from the aerosol-chemistry-climate model SOCOL-AERv1 to its updated version SOCOL-AERv2. In SOCOL-AERv2 several updates to the model have been implemented, e.g. interactive deposition schemes have been added, sulfate mass conservation has been ensured, and the tropospheric chemistry scheme has been extended. Results of both versions as well as of intermediate steps are compared with each other and to observational data in a sufficient manner. While the SOCOL-AERv2 results show with respect to stratospheric aerosol observations similar levels of agreement as SOCOL-

AERv1, the interactive deposition schemes in SOCOL-AERv2 lead to a much improved agreement with observed data. Overall SOCOL-AERv2 seems to be better suited to study the atmospheric sulfur cycle as its predecessor version.

This is a really nice paper and it was a pleasure to review it. In my opinion it is an excellent example how a paper in GMD should look like. The paper is well written, the abstract provides a concise and complete summary and the figures are nicely prepared. The different model steps are clearly outlined, scientifically sound and the applied methods and assumptions are valid. The different development steps are well documented and sufficiently explained. The reasoning behind the described procedure is clear. All model updates have a beneficial impact and make sense. Critical issues as deterioration from SOCOL-AERv1 to SOCOL-AERv2 for the extinction in the lower stratosphere and possible reasons for it are discussed as well.

I recommend the paper for publications after minor revisions

Specific comments:

Page 3, line 14: Please cite Zanchettin et al. (2016) as VolMIP reference

Page 4, line 11: If you give a reference for MEZON, please add also a reference of MA-ECHAM5 e.g. Giorgetta et al., (2006)

Page 17, line 14-18 wonder if the agreement between OPC data and model results for the larger particle sizes could be improved if one compare for Laramie not only the annual mean but also seasonal averages

Page 21, section 3.4.1.: As the authors mention, the precipitation fields in the model might not be correct. Thus, in order to avoid misinterpretation of the simulated station data, it might be worth to compare not only the total amount of the wet deposition flux but also a normalized one with respect precipitation (fraction of total wet deposition and precipitation).

Page 27, 1st para.: I suggest to discuss the weaknesses of the model not in the middle

of the paragraph but at the end. It might be worth to briefly discuss further possible improvements. A link to the ongoing ISA-MIP intercomparison (Timmreck et al., 2018) might also be useful as the multi-model approach (13 model groups have signed up incl. ETHZ) and the required detailed output diagnostics might be beneficial for further model improvements and a general assessment of the SOCOL-AERv2.

Figure 4: The figure is certainly very busy, but maybe it is possible to include also the uncertainty range of large particles

Figure 5: Please indicate also the uncertainty range of the model simulations

Figure 7: The accumulated lines are confusing and need a better explanation in the figure caption

Table 2: Please specify also the observational uncertainty range

References: Please revise the list carefully, often information about DOIs or pages are missing as for example for the two Deshler papers

References:

Giorgetta, M. A., Manzini, E., Roeckner, E., Esch, M., and Bengtsson, L.: Climatology and forcing of the quasi-biennial oscillationin the MAECHAM5 model, J. Climate, 19, 3882–3901, 2006.

Zanchettin, D. et al.: The Model Intercomparison Project on the climatic response to Volcanic forcing (VolMIP): experimental design and forcing input data for CMIP6, Geosci. Model Dev., 9, 2701-2719, https://doi.org/10.5194/gmd-9-2701-2016, 2016.

Timmreck, C. et al.: The Interactive Stratospheric Aerosol Model Intercomparison Project (ISA-MIP): motivation and experimental design, Geosci. Model Dev., 11, 2581-2608, https://doi.org/10.5194/gmd-11-2581-2018, 2018.

---

## Referee Comment (RC2) · Anonymous Referee #2 · 28 Jun 2019

This paper describes a new version of the aerosol-chemistry-climate model SOCOL-AERv2a, in particular very significant improvements to the tropospheric/stratospheric sulfur cycle. It is a bit of a textbook example for a GMD paper. The interesting results of the new version of the models are analyzed in detail but they are also systematically and thoroughly compared to the results obtained with the old version of the model including various sensitivity simulations. It is well written, clear and very helpful to the numerical modelling community working on atmospheric sulphate aerosols. I recommend publication. Nonetheless, the authors may wish to take on board the minor comments provided thereafter.

[Figure]

Abstract: There is something confusing in the abstract. It starts by recalling that there is a very good agreement between the stratospheric aerosol burden calculated by the old model version and a satellite-derived estimate (about 110 Gg S); I guess it is an old satellite estimate and it is for a volcanically quiescent aerosol burden. Later on, it is stated that that there is a good agreement between the stratospheric aerosol burden calculated by the new model version and a new satellite-derived estimate (about 160 Gg S). It is a bit confusing. Have the authors more confidence in the new satellite-derived estimate than in the old one? If that is the case (I have more confidence), why mention the old estimate in the abstract? I suggest to be selective and mention only the change in stratospheric aerosol burden from the old to the new model version and compare it to the new satellite-derived estimate.

P4, l16: I was wondering whether it would have been possible to run SOCOL by relaxing wind and T fields towards meteorological analyses. This would have partly removed sources of biases in global aerosol calculations, i.e. biases in transport or temperature-dependent processes such as condensation/evaporation as the top of the aerosol layer etc. . . I think that a more realistic transport and temperatures would help to confine the origins of biases to aerosol emissions, microphysical processes, and deposition.

P4, l27: I have some doubts about the realism of nucleation calculations (Vehkamäki et al., 2002) based on grid-box temperature and humidity in a global model. One would expect nucleation, a highly non-linear process, to occur in environmental extremes at sub-grid scales. Having said that, all models use this approach without accounting for sub-grid scale variability.

p20 ,l13-16: I agree that a too large cross-tropopause flux is a common bias in global models. On the top of convective transport and aerosol-specific heterogeneous processes, I think a big problem is the model resolution, in particular vertical. The vertical gradients in chemical fields are very steep at the tropopause, difficult to avoid artificial diffusion when the vertical resolution is of the order of a km, typical in global models. For instance, cross-tropopause ozone fluxes tend to be overestimated in global

models.

---

## Author Comment (AC1) · 18 Jul 2019

**Author response to both referees' comments on "Improved tropospheric and stratospheric sulfur cycle in the aerosol-chemistry-climate model SOCOL-AERv2"**

We thank both reviewers for taking the time to read the manuscript and for their helpful comments about the manuscript. We have taken these comments into account and present our responses below, with reviewer comments in blue and author responses in black.

**Response to C. Timmreck**

Page 3, line 14: Please cite Zanchettin et al. (2016) as VolMIP reference
The reference to Zanchettin et al. (2016) has been added.

Page 4, line 11: If you give a reference for MEZON, please add also a reference of MA-ECHAM5 e.g. Giorgetta et al., (2006)
References for Roeckner et al. (2003) and Giorgetta et al. (2006) have been added to this paper.

Page 17, line 14-18 wonder if the agreement between OPC data and model results for the larger particle sizes could be improved if one compare for Laramie not only the annual mean but also seasonal averages
We have compared the seasonal averages of the OPC data with SOCOL-AERv2, shown below. The model overestimates the number densities of larger size bins in all seasons, with the exception of boreal autumn where this bias is slightly reduced. However, the dynamical factors causing the reduced bias in boreal autumn are not clear to us. The measurement lines seem to shift more than the model lines between other seasons and boreal autumn.

[Figure]

 As the authors mention, the precipitation fields in the model might not be correct. Thus, in order to avoid misinterpretation of the simulated station data, it might be worth to compare not only the total amount of the wet deposition flux but also a normalized one with respect precipitation (fraction of total wet deposition and precipitation).

We appreciate these suggestions and we have tried to plot the normalized quantity. However, this quantity would not make sense for SOCOL-AERv1 since the model's deposition is not linked to precipitation fields. It is difficult to interpret deviations in the normalized deposition, since it is not a commonly used quantity in deposition studies. We have added a plot to the supplementary material that shows the relationship between sulfate deposition biases with precipitation biases (shown below, with the Spearman correlation coefficient listed). This helps visualize the correlation that was mentioned in the manuscript on page 22, line 12.

[Figure]

 I suggest to discuss the weaknesses of the model not in the middle of the paragraph but at the end. It might be worth to briefly discuss further possible improvements. A link to the ongoing ISA-MIP intercomparison (Timmreck et al., 2018) might also be useful as the multi-model approach (13 model groups have signed up incl. ETHZ) and the required detailed output diagnostics might be beneficial for further model improvements and a general assessment of the SOCOL-AERv2.

The paragraph was reordered so that the weaknesses of SOCOL-AERv2 appear at the end of this paragraph. We also added an outlook sentence, linking to ISA-MIP: "Comparison with other models in the background experiments of the Interactive Stratospheric Aerosol Model Intercomparison Project (ISA-MIP) (Timmreck et al., 2018) may help to identify the cause of SOCOL-AERv2's overestimated lower stratospheric aerosol."

 The figure is certainly very busy, but maybe it is possible to include also the uncertainty range of large particles

Please note that the box plots shown on Figure 4 are the range in OPC measurements, not the uncertainty range of the model simulations. Below we show a plot including the variability in model monthly means: the model median is shown as a solid line and the 25[th] and 75[th] percentile values are shown as dashed lines. The simulated variability is in general smaller than the observed variability. We decided to not

include these uncertainty ranges on the figure in the manuscript, since as you mention the figure is already quite busy. We did enlarge Figure 4 in the vertical direction in the revised manuscript, to make it more clear that the observational box plots are not shading for the modelled variability.

[Figure]

Figure 5: Please indicate also the uncertainty range of the model simulations

Figure 5 (shown below) has been updated to include the interquartile range of the monthly means in the model simulations.

[Figure]

Figure 7: The accumulated lines are confusing and need a better explanation in the figure caption

Thank you for this feedback. We have amended the caption of the legend to read: "Orange and grey lines represent the accumulated (acc) globally integrated deposition of sulfur emitted from Pinatubo. The accumulated deposition from Pinatubo is calculated from simulations where all sulfur sources other than Pinatubo are turned off."

Table 2: Please specify also the observational uncertainty range

We have included the uncertainty range of the WMO assessment using the standard deviation of the models in the assessment. The SAGE-$3\lambda$ and SAGE-$4\lambda$ standard deviations were also included in Table 2 and the text, using the variability in the derived monthly mean burdens.

References: Please revise the list carefully, often information about DOIs or pages are missing as for example for the two Deshler papers

DOIs have been added to all references, where available. Page numbers were also added to the two Deshler papers.

**Response to Referee #2**

Abstract: There is something confusing in the abstract. It starts by recalling that there is a very good agreement between the stratospheric aerosol burden calculated by the old model version and a satellite-derived estimate (about 110 Gg S); I guess it is an old satellite estimate and it is for a volcanically quiescent aerosol burden. Later on, it is stated that that there is a good agreement between the stratospheric aerosol burden calculated by the new model version and a new satellite-derived estimate (about 160 Gg S). It is a bit confusing. Have the authors more confidence in the new satellite-derived estimate than in the old one? If that is the case (I have more confidence), why mention the old estimate in the abstract? I suggest to be selective and mention only the change in stratospheric aerosol burden from the

old to the new model version and compare it to the new satellite-derived estimate.

We agree that this can be misleading to mention two satellite-derived estimates in the abstract, even though it is discussed in depth within the paper itself. We have edited the abstract accordingly, only mentioning the increase in burden from SOCOL-AERv1 to v2 and the agreement with SAGE-3$\lambda$.

P4, l16: I was wondering whether it would have been possible to run SOCOL by relaxing wind and T fields towards meteorological analyses. This would have partly removed sources of biases in global aerosol calculations, i.e. biases in transport or temperature-dependent processes such as condensation/evaporation as the top of the aerosol layer etc... I think that a more realistic transport and temperatures would help to confine the origins of biases to aerosol emissions, microphysical processes, and deposition.

Thank you for this suggestion. In the study, we followed the experimental design of Sheng et al. (2015) for background sulfur conditions, to best compare the two versions of the model. Nevertheless, to answer this question we have conducted an additional SOCOL-AERv2 2000–2010 transient run with specified dynamics, relaxing wind and T fields to ERA-Interim reanalysis data (Dee et al., 2011). The nudged version actually slightly degrades the agreement of SOCOL-AERv2 with precipitation measurements from the wet deposition networks compared to the free running version, possibly due to the effect that nudging has on convective precipitation. To compare with the background sulfur observations from the paper, we average nudged and free-running model results between 2000–2004. The nudged and free running runs are very similar when comparing to OPC measurements. The nudged version shows higher amounts of $SO_2$ in the tropical upper troposphere, which is farther away from the measurements in Figure 5. For the comparison with SAGE extinctions, the model's stratospheric high bias at the Equator between 20 and 30 km and at 45° N below 20 km remains in the nudged version (see figure below). The nudged version does fall closer to the measured extinction values between 30 and 37 km at the Equator. This may be related to the improved representation of dynamics in this region. However, the main aspects that we discuss in the paper are unchanged, namely the stratospheric bias below 30 km.

[Figure]

P4, l27: I have some doubts about the realism of nucleation calculations (Vehkamäki et al., 2002) based on grid-box temperature and humidity in a global model. One would expect nucleation, a highly non-linear process, to occur in environmental extremes at sub-grid scales. Having said that, all models use this approach without accounting for sub-grid scale variability.

We agree that sub-grid scale variability would have an impact on nucleation. We do think that this problem would be more relevant to modelling stratospheric aerosols after volcanic eruptions or geo-engineering than in the background sulfur cycle. There is definitely interest in coupling global aerosol models with small-scale Lagrangian plume dispersion models (for example, suggested by Vattioni et al. (2019)), which may help account for this sub-grid scale variability. However, as mentioned by the reviewer, the current state-of-the-art aerosol models use this parametrization with grid-scale temperatures and humidities.

p20 ,l13-16: I agree that a too large cross-tropopause flux is a common bias in global models. On the top of convective transport and aerosol-specific heterogeneous processes, I think a big problem is the model resolution, in particular vertical. The vertical gradients in chemical fields are very steep at the tropopause, difficult to avoid artificial diffusion when the vertical resolution is of the order of a km, typical in global models. For instance, cross-tropopause ozone fluxes tend to be overestimated in global models.

This is an important point that the reviewer addresses. We have added a sentence in this discussion: "Numerical diffusion may further enhance the $SO_2$ cross-tropopause transport, due to the strong vertical gradient at the tropopause".

[revised manuscript text omitted]